# Impact of $HO_2/RO_2$ ratio on highly oxygenated α-pinene photooxidation products and secondary organic aerosol formation potential

Yarê Baker[1], Sungah Kang[1], Hui Wang[1], Rongrong Wu[1], Jian Xu[1], Annika Zanders[1], Quanfu He[1], Thorsten Hohaus[1], Till Ziehm[1], Veronica Geretti[2], Thomas J. Bannan[3], Simon P. O'Meara[3,4], Aristeidis Voliotis[3], Mattias Hallquist[2], Gordon McFiggans[3], Sören R. Zorn[1], Andreas Wahner[1], and Thomas F. Mentel[1]

[1]Institute for Energy and Climate Research, IEK-8, Forschungszentrum Jülich, 52425 Jülich, Germany
[2]Atmospheric Science, Dept. of Chemistry, University of Gothenburg, Gothenburg, 412 96, Sweden
[3]Department for Earth and Environmental Sciences, University of Manchester, Manchester, M13 9PL, UK
[4]National Centre for Atmospheric Science, University of Manchester, Manchester, M13 9PL, UK

*Correspondence to*: Thomas F. Mentel (t.mentel@fz-juelich.de)

**Abstract.** Highly oxygenated molecules (HOM) from the atmospheric oxidation of biogenic volatile organic compounds are important contributors to secondary organic aerosol (SOA). Organic peroxy radicals ($RO_2$) and hydroperoxy radicals ($HO_2$) are key species influencing the HOM product distribution. In laboratory studies experimental requirements often result in overemphasis of $RO_2$ cross-reactions compared to reactions of $RO_2$ with $HO_2$. We analyzed the photochemical formation of HOMs from α-pinene and their potential to contribute to SOA formation under high (≈1/1) and low (≈1/100) $HO_2/RO_2$ conditions. As $HO_2/RO_2 > 1$ is prevalent in the daytime atmosphere, sufficiently high $HO_2/RO_2$ is crucial to mimic atmospheric conditions and to prevent biases by low $HO_2/RO_2$ on the HOM product distribution and thus SOA yield. Experiments were performed under steady-state conditions in the new, continuously stirred tank reactor SAPHIR-STAR at Forschungszentrum Jülich. The $HO_2/RO_2$ ratio was increased by adding CO, while keeping the OH concentration constant. We determined the HOM's SOA formation potential, considering their fraction remaining in the gas phase after seeding with $(NH_4)_2SO_4$ aerosol. Increase of $HO_2/RO_2$ led to a reduction in SOA formation potential, with the main driver being a ≈60% reduction in HOM-accretion products. We also observed a shift in HOM-monomer functionalization from carbonyl to hydroperoxide groups. We determined a reduction of the HOM's SOA formation potential by ≈30% at $HO_2/RO_2$≈1/1. Particle phase observations measured an about according decrease in SOA mass and yield. Our study showed that too low $HO_2/RO_2$ ratios compared to the atmosphere can lead to an overestimation of SOA yields.

## Introduction

In the atmosphere highly oxidized products from the oxidation of biogenic or anthropogenic volatile organic compounds (VOCs) are an important source of secondary organic aerosol (SOA) (Roldin et al., 2019; Mohr et al., 2019). SOA is an important contributor to the overall ambient aerosol and of interest because of its impact on climate, visibility, and human health (Hallquist et al., 2009).

Recently, many studies (Pullinen et al., 2020; Berndt et al., 2016; Bianchi et al., 2017) have focused on understanding the oxidation pathways of VOCs that yield highly oxygenated molecules (HOMs), as these are expected to be of low enough volatility to condense into the particle phase. One important tool for the investigation of VOC degradation and SOA formation is the utilization of experiments in atmospheric simulation chambers (Hidy, 2019). Such experiments have also helped to elucidate key processes in the HOM formation, i.e. the process of autoxidation.

After an initial oxidant attack and the formation of a peroxy radical ($RO_2$), autoxidation adds oxygen to the molecule via an internal H-shift to the peroxy group, forming a hydroxy peroxide group and an alkyl radical, to which $O_2$ immediately adds, reestablishing the peroxy functionality. This process can be repeated multiple times yielding almost instantaneously highly oxygenated peroxy radicals (HOM-$RO_2$) which are terminated to a series of HOM closed-shell products (Bianchi et al., 2019; Ehn et al., 2014; Crounse et al., 2013).

Chamber studies often work with a singular compound and operate at higher precursor concentrations than those observed in the atmosphere for experimental reasons. These experiments cannot represent the complex mixture of VOCs and oxidized VOCs present in the atmosphere (McFiggans et al., 2019). Higher precursor concentrations can lead per se to higher SOA yields than observed in the atmosphere (a well characterized phenomenon (see Henry et al. (2012), Shilling et al. (2009)) and to a general preference of higher order processes which may not be important in the atmosphere. One example is that chamber studies tend to overestimate the role of cross reactions between organic peroxy radicals ($RO_2$) owing to high precursor concentrations of a single VOC. In chambers, reactions of HOM-$RO_2$ with other organic peroxy radicals terminate the autoxidation chain, leading typically to multifunctional carbonyl and alcohol compounds. In comparison, in the atmosphere termination by $HO_2$ is more likely, leading to multifunctional hydroperoxides. In presence of sufficient NO, termination to multifunctional organic nitrates may be more important (Schervish and Donahue, 2021).

Another possible termination reaction of HOM-$RO_2$ with HOM-$RO_2$ and less oxidized $RO_2$ leads to the formation of accretion products, which are expected to be extremely low volatile organic compounds (ELVOCs) and are therefore expected to contribute to new particle formation and SOA formation (Ehn et al., 2014; Berndt et al., 2018). Schervish and Donahue (2021) raised awareness that chamber studies could overestimate the SOA formation potential from the oxidation of terpenes such as α-pinene compared to the atmosphere, because of missing $HO_2$ and small $RO_2$ (e.g. $CH_3O_2$), which favors accretion product formation.

In chamber studies the use of higher VOC concentrations is often an unavoidable necessity either to match the sensitivity of the analytical instrumentation or to overcome chamber related effects. The question remains, how can conditions dictated by the chamber be steered towards more realistic chemical pathways and higher atmospheric relevance?

In this study we address this overestimated importance of peroxy radical cross reactions. We studied the photooxidation of α-pinene in a series of steady-state experiments in the newly built continuously stirred tank reactor SAPHIR-STAR (a modernized version of JPAC, see Mentel et al. (2009)).

In these experiments, after an initial α-pinene photooxidation phase as a reference, CO was added to the oxidation system to represent small, oxidized VOCs in the atmosphere that can produce $HO_2$ by reaction with OH (compare Schervish and Donahue (2021)). Presence of CO shifts the $HO_2$ to $RO_2$ ratio, increasing the importance of the $RO_2$ termination with $HO_2$. However, McFiggans et al. (2019) showed that one limiting factor in mixture experiments is oxidant scavenging: the products and their yields in mixed systems change, because there is less OH available to the individual VOC. Thus, after the CO addition the OH production in the chamber was increased to compensate for the oxidant scavenging. The OH levels in the system before and after the CO addition were approximately the same, keeping the α-pinene OH turnover, as well as the primary peroxy radical production approximately constant.

Furthermore, the addition of seed particles ($(NH_4)_2SO_4$) allowed us to observe the condensation behavior of the HOM-products and to compare our gas phase observations directly with particulate phase measurements of the condensed organic mass.

In this study we will address two central questions: How does the shift in $HO_2/RO_2$ impact the oxidation mechanism of α-pinene, especially the HOM formation pathway? And what is the subsequent impact on the SOA formation potential of the α-pinene photooxidation system? As the central analysis tool, we will use high resolution time of flight mass spectrometry with chemical ionization (HR-TOF-CIMS).

# 1 Methods

## 1.1 Generic α-pinene HOM peroxy radical chemistry

The chemical mechanistic information for the basic oxidation scheme of α-pinene was taken from the Master Chemical Mechanism MCM v3.3.1 (Jenkin et al., 1997; Saunders et al., 2003) (http://mcm.york.ac.uk). The main peroxy radicals expected from α-pinene photooxidation are $C_{10}H_{17}O_x$ and $C_{10}H_{15}O_x$. $C_{10}H_{17}O_x$ is formed by the addition of OH to α-pinene, followed by $O_2$ (starting $RO_2$: $C_{10}H_{17}O_3$) (MCM v3.3.1 (Jenkin et al., 1997; Saunders et al., 2003)). Studies showed that the autoxidation can start from $C_{10}H_{17}O_3$ with the four-member ring in α-pinene opened (Berndt, 2021; Xu et al., 2019).

For $C_{10}H_{15}O_x$ the autoxidation chain is assumed to start with $C_{10}H_{15}O_4$, which can be formed directly from ozonolysis via the vinyl hydroperoxide path (Johnson and Marston, 2008; Iyer et al., 2021) or via H-abstraction from first-generation oxidation

products such as pinonaldehyde ($C_{10}H_{16}O_2$). (MCM v3.3.1 (Jenkin et al., 1997; Saunders et al., 2003; Fantechi et al., 2002). A recent study suggests direct H-abstraction by OH from α-pinene (Shen et al., 2022).

The autoxidation process is rapid with H-shift rates of about 0.01 - 0.1 $s^{-1}$ and faster (Piletic and Kleindienst, 2022; Berndt, 2021; Xu et al., 2019; Vereecken et al., 2007). Once the autoxidation process starts it quickly adds more oxygen to the molecule, until the difficulty in abstracting remaining H-atoms slows down the reaction sufficiently such that termination reactions can compete. In the absence of $NO_x$, the peroxy radicals have two major bimolecular termination channels: the reaction with another $RO_2$ or with $HO_2$. A third pathway is the intramolecular termination (Rissanen et al., 2014).

Based on the considerations above, we apply a simplified generic reaction scheme to analyze our observations. **Figure 1** shows an overview of the reaction pathways for the main peroxy radical families in the α-pinene photooxidation and the resulting product groups and families. The compounds can be separated into four classes; peroxy radicals (HOM-$RO_2$), monomers (HOM-Mon), accretion products (HOM-Acc) and fragments (HOM-Frag). The HOM-$RO_2$ class consists of all detected HOM-$RO_2$, with special focus on the analysis of the $C_{10}$ HOM-$RO_2$ family. The HOM-Mon class contains the closed-shell HOM-$C_{10}$ products. The compounds in the fragment class contain less than ten carbon atoms, while all HOM-Acc compounds contain more than ten carbon atoms. The compound classes are further divided into groups and families. Here, the term group is used for compounds with the same carbon number, while a family contains all compounds with the same carbon and hydrogen number but a varying oxygen number.

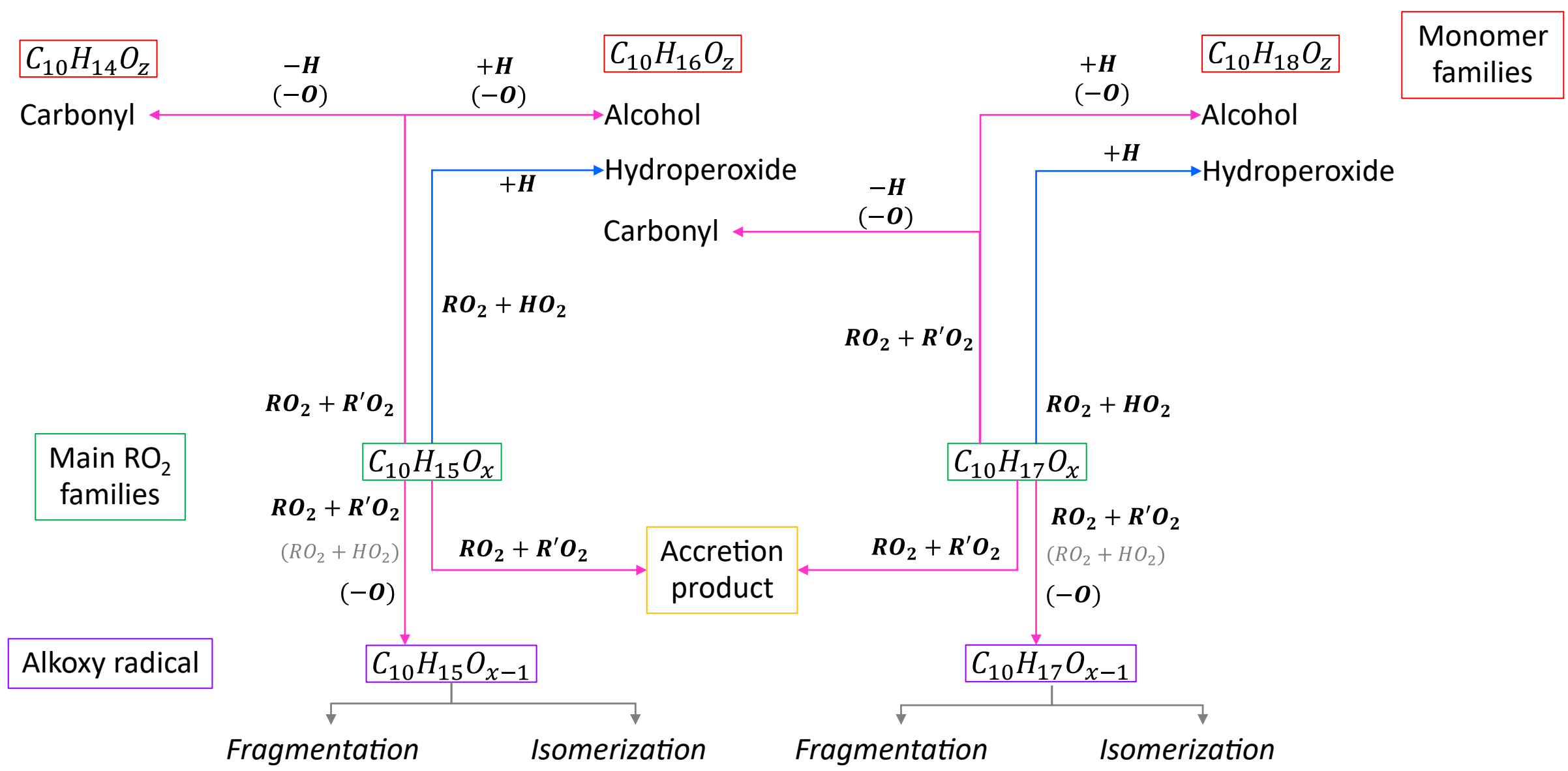

**Figure 1: Overview of important reaction pathways of α-pinene $RO_2$ with other $RO_2$ and $HO_2$.**

The termination of $RO_2$ with $HO_2$ will lead to hydroperoxide formation:

$$RO_2+HO_2 \rightarrow ROOH + O_2 \qquad \textbf{\textit{(R1)}}$$

In the case of $C_{10}H_{15}O_x$, reaction ***(R1)*** will lead to multifunctional $C_{10}H_{16}O_z$ hydroperoxides (wherein the notation "hydroperoxides" or "carbonyls", "alcohols" etc. here and in the following relates to the functionality of the group formed by the termination reaction). For $C_{10}H_{17}O_x$ it will lead to the formation of $C_{10}H_{18}O_z$ hydroperoxides. The termination via $RO_2$+$RO_2$ can either result in the formation of accretion products or in the formation of carbonyls and alcohols. For the accretion product formation, it is assumed that the two $RO_2$ chemically bond eliminating $O_2$ from the molecule:

$$RO_2+R'O_2 \rightarrow R\text{-}O\text{-}O\text{-}R'+O_2 \qquad \textbf{\textit{(R2)}}$$

Recombination reactions of the main peroxy radical families $C_{10}H_{15}O_x$ and $C_{10}H_{17}O_x$ lead to the product families $C_{20}H_{30}O_z$ (combination of two $C_{10}H_{15}O_x$), $C_{20}H_{32}O_z$ (combination of $C_{10}H_{15}O_x$ and $C_{10}H_{17}O_x$), and $C_{20}H_{34}O_z$ (combination of two $C_{10}H_{17}O_x$).

However, due to reactions with smaller peroxy radicals, HOM-Acc families with smaller carbon and hydrogen numbers are also observed. Indeed, one reason why the $RO_2$+R'$O_2$ termination is expected to affect the SOA formation potential is the formation of accretion products by scavenging of less oxidized and smaller $RO_2$ by HOM-$RO_2$. Thus, the smaller $RO_2$ will also contribute to the SOA mass which would otherwise not be the case. For the HOM-$RO_2$ itself, it is expected that they contribute to SOA formation independently of the termination pathway, due to the low volatility of its expected termination products (Pullinen et al., 2020; McFiggans et al., 2019).

The second $RO_2$+R'$O_2$ termination pathway is the formation of a carbonyl and alcohol compound:

$$RO_2+R'O_2 \rightarrow R\text{-}OH + R'{=}O + O_2 \qquad \textbf{\textit{(R3)}}$$

In this reaction both radicals lose an oxygen atom, and a hydrogen atom is transferred to the $RO_2$ forming the alcohol termination group. Preferences of $RO_2$ to form an alcohol or carbonyl compound are possible for individual reactions, but statistically carbonyl and alcohols should be formed with the same fractions. Since mass spectrometry can only determine formula composition, we cannot distinguish alcohols and hydroperoxides, which arise from $RO_2$ differing by one O atom. Therefore, details of balance of alcohol and carbonyl formation cannot be detected.

However, the formula composition can help to differentiate certain formation pathways. The $C_{10}H_{14}O_z$ family contains only carbonyl formed from a $C_{10}H_{15}O_x$ $RO_2$ while the alcohol will be part of the $C_{10}H_{16}O_z$ family. The $C_{10}H_{16}O_z$ family also contains the carbonyl produced from the $RO_2$+R'$O_2$ monomer termination of $C_{10}H_{17}O_x$, while the alcohol from this $RO_2$ family will be found in the $C_{10}H_{18}O_z$ family. So, from a diagnostic point of view, $C_{10}H_{14}O_z$ as well as $C_{10}H_{18}O_z$ are uniquely related to a precursor radical family.

The classification of the formation pathways of the monomers is helpful to analyze the effect of the $HO_2/RO_2$ ratio shift in the experiments. Considering the termination pathways, a decrease in the $C_{10}H_{14}O_z$ family and an increase of the $C_{10}H_{18}O_z$ family is expected with increasing $HO_2/RO_2$ because of increasing termination by $HO_2$ and decreasing termination by $RO_2$. In case of $C_{10}H_{18}O_z$ the increase of hydroperoxides is partially compensated by a decrease of the alcohol channel. For $C_{10}H_{16}O_z$ the situation is more complicated as it contains contributions from all termination pathways.

Besides closed-shell products, HOM-$RO_2$ can also form alkoxy radicals (HOM-RO). In general, alkoxy radicals (RO) are important intermediates in the oxidation scheme of organics and are formed via ***(R4)*** and probably also via ***(R5)*** for specific $RO_2$ (Jenkin et al., 2019):

$$RO_2 + R'O_2 \rightarrow RO + R'O + O_2 \qquad \textbf{\textit{(R4)}}$$

$$RO_2 + HO_2 \rightarrow RO + OH + O_2 \qquad \textbf{\textit{(R5)}}$$

In reaction ***(R5)*** OH will be formed. The importance of reaction ***(R5)*** compared to reaction ***(R1)*** is still unclear in the literature, but functionalization of the $RO_2$ close to the peroxy functionality possibly enables this reaction (Iyer et al., 2018; Eddingsaas et al., 2012; Hasson et al., 2005; Jenkin et al., 2019). If reaction ***(R5)*** is of negligible importance, the reaction scheme will simplify and the effect of increased $HO_2/RO_2$ is easier to diagnose.

We are interested in the importance of alkoxy radical formation as (HOM)-RO tend to fragment, leading to the formation of smaller products (Vereecken et al., 2007). In the context of SOA formation, these fragments are less likely to contribute to SOA mass because of their higher volatility. Since alkoxy radicals are too unstable to be detected directly we use two diagnosis tools to judge the importance of HOM-RO. Firstly, HOM-RO fragmentation can lead to HOM-$RO_2$ with less than 10 carbon atoms which may also continue the autoxidation chain. Therefore, the abundance of HOM with less than 10 carbon atoms (HOM-Frag) indicates the importance of alkoxy steps. Secondly, with increasing functionalization, H-shifts retaining the carbon backbone become more likely (Vereecken et al., 2007) which will lead to a next generation of $C_{10}$-HOM-$RO_2$. Such alkoxy peroxy steps can continue the autoxidation chain (Mentel et al., 2015). Interestingly, by coupling of an alkoxy and a peroxy step, the parity of the number of oxygen atoms in the HOM-$RO_2$ changes, while in pure autoxidation steps the oxygen parity remains the same. Therefore, a parity change of the oxygen number can be used as an indication of alkoxy step abundance (Kang, 2021).

In summary we will use the changes in contribution and relative signal of the different families and classes to judge the impact of shifting from low to high $HO_2/RO_2$ on the α-pinene photooxidation pathway.

## 1.2 Control of α-pinene OH turnover

One important concept of the conducted experiments is the constant OH availability to α-pinene in the mixtures with CO to avoid effects of oxidant scavenging (McFiggans et al., 2019). Therefore, after each change in the $HO_2/RO_2$ regime by CO addition, the OH level was readjusted to yield the same α-pinene OH turnover and compensate for the OH consumed by CO. This OH adjustment ensures that the primary α-pinene chemistry was kept the same and enables a direct comparison.

However, since experiments could only be performed at *about* the same OH levels, a normalization by the actual α-pinene OH turnover is applied to the data. This compensates for the slight experimental imperfections and enables better comparison of experiment series with different boundary conditions. The turnover in steady state is given in **Eq. (1)**. Here the subscript "SS" denotes steady state condition for the concentrations of α-pinene and OH, $k_{OH}$ is the α-pinene OH reaction rate constant.

$$turnover_{apinene+OH} = k_{OH} * [\alpha\text{-}pinene]_{SS} * [OH]_{SS} \quad \textbf{(1)}$$

This normalization also directly shows the yield of certain oxidation product or product group per α-pinene consumed by OH.

## 1.3 Derivation of effect on condensable mass from gas-phase measurement

A simple proxy for the condensable mass from HOM products can be calculated from the steady-state HOM-signals measured by the $NO_3$-CIMS, assuming condensation for all low volatility HOM-compounds and no back evaporation into the gas phase. To only take low volatility products into account we used all detected formula compositions with M> 230 g $mol^{-1}$ and weighted them with their molar mass. The reasoning behind this threshold can be found in **Sect. 4.4**. All contributions were summed up and normalized with the α-pinene OH turnover for the comparison between the low and high $HO_2/RO_2$ cases (**Eq. (2)**).

$$mass\ weighted\ signal\ sum = \frac{\sum_{i=0}^{i} S_i * M_i}{turnover_{apinene+OH}} \quad \textbf{(2)}$$

We also estimated the expected SOA mass formed using the calibration factor obtained for sulfuric acid for our $NO_3$-CIMS instrument in a calibration setup (see supplement **Sect. S1**). From this we calculated an upper boundary concentration of detected HOM-compounds in the gas phase under the assumption that sulfuric acid clusters with nitrate at the collision limit, yielding maximum sensitivity (a common approach, see for example Ehn et al. (2014), Pullinen et al. (2020)).
The calculated gas phase concentration was then used in the steady state equation describing the relationship between gas and particle phase concentrations of a single compound *i* shown in **Eq. (3)**.

$$m_{i,seed}(p) = \frac{m_{i,seed}(g) * k_{cond,i}}{k_{particleLoss} + k_{evap,i}} \quad \textbf{(3)}$$

**Equation (3)** shows that the steady state particle phase (mass) concentration $m_{i,seed}(p)$ of compound $i$ in presence of seed in the chamber is only dependent on the steady state gas phase concentration $m_{i,seed}(g)$, the condensation rate and evaporation rate constants $k_{cond,i}$, $k_{evap,i}$ of $i$ (to and from the particles) and the particle loss rate constant $k_{particleLoss}$ in the chamber. The condensation rate can be calculated (see supplement **Sect. S6**), and the particle loss rate constant was measured by observation of the particle loss in the chamber after ending the seed addition (details in the supplement **Sect. S2**). The evaporation rate was assumed to be negligible for the investigated HOM-compounds.

For the SOA yield calculation, we calculate a corrected organic mass $m_{SOA}$ from the organic mass $m_{AMS}$ measured by aerosol mass spectrometry (AMS) and the fraction expected to be lost on the seed particles compared to the overall loss on particles and chamber wall as shown in **Eq. (4)** (McFiggans et al., 2019).

$$m_{SOA} = m_{AMS} * \frac{k_{cond} + k_{wall}}{k_{cond}} \quad \mathbf{(4)}$$

In **Eq. (4)** we use the condensation rate constant $k_{cond}$ calculated for one major HOM-product ($C_{10}H_{16}O_7$) and the average HOM-Mon wall loss rate $k_{wall}$ which was determined by switching off the UVC light and observing the decay of photooxidation products in the $NO_3$-CIMS. The wall loss determination, as well as SOA mass correction were described before in Sarrafzadeh et al. (2016) and McFiggans et al. (2019).

# 2 Experimental methods

## 2.1 Chamber setup

Experiments were conducted in the Jülich SAPHIR STAR chamber, which is the modern successor of the JPAC setup (Mentel et al., 2009). The basic concepts are the same as in JPAC, but each parameter is set, controlled, and monitored in a program. The chamber was operated as a continuously stirred tank reactor. It is a borosilicate glass cylinder (l=2.5 m, d=1 m) with a volume of 2000 L and all equipment inside the chamber is either glass or glass coated steel (SilcoTek GmbH).

With an inflow of 32 L min$^{-1}$, the residence time in the chamber was approximately 63 minutes with a fan ensuring mixing within minutes. In contrast to the JPAC chamber, the stirring is conducted perpendicular to the cylinder axis, as opposed to coaxial. Chamber inflow is split into two humidified clean air flows (mixed from $N_2$ and $O_2$) of about equal volume, one with added oxidant (here $O_3$), the other with added VOC and other trace gases (here α-pinene and CO). All experiments were performed at a relative humidity of 50 % and at 20 °C. Temperature stability is ensured by the climate-controlled surrounding of the chamber.

α-Pinene (≥99 % purity, Sigma-Aldrich Merck KGaA) was introduced via liquid injection with a syringe pump (Fusion 4000, CHEMYX Inc.) into a heated glass bulb and flushed by a stream of 1 L min$^{-1}$ into the chamber. CO was added from a

gas bottle (10% CO in $N_2$, Messer SE & Co. KGaA). Ozone was directly produced photolytically before injection with a self-built ozone generator.

OH is produced in the chamber by ozone photolysis using two UV-C lamps with a wavelength of 254 nm and subsequent reaction of $O(^1D)$ with water vapor. The lamps are mounted in closed quartz cylinders in the middle of the chamber, vertically to the cylinder axis and light intensity can be varied with a movable shielding installed around the lamps. The shielding allows an exact percentage of the lamp to be covered, thus controlling the amount of OH produced in the chamber.

The OH radical concentration after CO addition was adjusted by setting the shielding of the UVC lamps and a slight adjustment of $O_3$ inflow. The applied $J(O^1D)$ values in different phases were calculated to be in the range of $0.8 \cdot 10^{-3}$ to $2.4 \cdot 10^{-3}$ $s^{-1}$.

In some of the experiments, ammonium sulfate (≥99 % purity, Merck KGaA) seed particles were added to the system to provide a surface for the condensation of organic material. The aerosol was produced with a modified TSI atomizer (Model 3076, TSI GmbH) and dried to 50% relative humidity.

VOC concentrations in the chamber were measured using proton-transfer-reaction mass spectrometry (PTR-TOF-MS; Ionicon GmbH). $CO_2$, CO, $H_2O$ (G2401 Cavity Ringdown Spectrometer, Picarro Inc.), NO, $NO_x$ (NCLD899, Eco Physics GmbH with a home-built photolytic converter) and $O_3$ (O342e, Envea GmbH) were additionally monitored. Particle distribution and concentration were measured with a condensation particle counter (CPC, Model 3788, TSI GmbH) and a scanning mobility particle sizer (SMPS; Model 3080, TSI GmbH) with a CPC (Model 3788, TSI GmbH). The aerosol composition was measured with a high-resolution aerosol mass spectrometer (HR-TOF AMS; Aerodyne Inc.).

In all experiments, VOC, $O_3$, and SMPS+CPC sampling switched between inlet and outlet of the chamber to measure the input concentrations as well as the concentrations in the reactor. The flow control system of the chamber adapts to these switches so that the inflow into the chamber stays constant.

All results discussed here were observed under steady-state conditions when all parameters were constant. For each steady state, the OH concentration was calculated from the decay of α-pinene as described by Kiendler-Scharr et al. (2009). **Equation (5)** is derived from the mass balance of α-pinene at steady state. The steady state OH concentration $[OH]_{SS}$ depends on the amount of α-pinene consumed by reaction with OH and the reaction with $O_3$, as well as the flush out.

$$[OH]_{SS} = \frac{\frac{F}{V} * \frac{[VOC]_{in} - [VOC]_{SS}}{[VOC]_{SS}} - k_{O3} * [O_3]_{SS}}{k_{OH}} \quad \textbf{(5)}$$

Here, F is the total flow and V the volume of the chamber. The subscript "SS" indicates steady-state concentrations, while $[VOC]_{in}$ represents the α-pinene concentration entering the chamber. $k_{o3}$ and $k_{OH}$ represent the reaction rate constants of α-pinene with the corresponding oxidant. We applied rate coefficients of $k_{o3}=5.36 \cdot 10^{-11}$ $cm^3 \cdot s^{-1}$ (Atkinson and Arey, 2003) and

$k_{OH}$=9.25·10$^{-17}$ cm$^3$·s$^{-1}$ (Cox et al., 2020) at 20 °C. The uncertainty of the OH calculation was estimated as 20 % by Wildt et al. (2014).

## 2.2 Experiment conditions

An overview of the experiments and their boundary conditions can be found in **Table 1**. Four experiments were performed in total. In two of the experiments ammonium sulfate seeds were added leading to a total particle surface in the chamber on the order of 8·10$^{-4}$ m$^2$ m$^{-3}$ and organic loadings of about 3 ug m$^{-3}$ in the photooxidation stage. The *Exp2* experiment is a consecutive combination of a seeded, followed by a non-seeded experiment to provide direct insight into the effect of seed presence on the system.

As the OH radical is produced by photolysis of ozone and α-pinene reacts with ozone, it is important to know the relative contribution of the α-pinene consumption by OH and by $O_3$. This is achieved by comparing the turnover of α-pinene with OH and $O_3$ respectively. The results can be found in **Table 1**. The listed results are for the low $HO_2/RO_2$ conditions, but nearly identical values were reached after the $HO_2/RO_2$ shift.

**Table 1. Overview of experimental conditions**

| **Name** | **Experiment description** | **$[VOC]_{in}$** | **$[CO]_{in}$** | **$[OH]_{ss}$ at low $HO_2/RO_2$** | **Contribution of OH to turnover at low $HO_2/RO_2$** | **Particle surface at low $HO_2/RO_2$** | **Organic mass concentration at low $HO_2/RO_2$** |
|---|---|---|---|---|---|---|---|
| *Exp1* | *pure gas phase (1)* | 10 ppbv | 2.5 ppmv | 4.1E+6 cm$^{-3}$ | 80 % | - | - |
| *Exp2.1* | *seeded (1)* | 10 ppbv | 2.5 ppmv | 1.0E+7 cm$^{-3}$ | 91 % | 8.7E-4 m$^2$ m$^{-3}$ | 3.4 µg m$^{-3}$ |
| *Exp2.2* | *unseeded (2)* | 10 ppbv | 2.5 ppmv | 1.3E+7 cm$^{-3}$ | 93 % | - | - |
| *Exp3* | *seeded (2)* | 10 ppbv | 2.5 ppmv | 1.4E+7 cm$^{-3}$ | 79 % | 6.8E-4 m$^2$ m$^{-3}$ | 2.7 µg m$^{-3}$ |

## 2.3 Experimental procedure

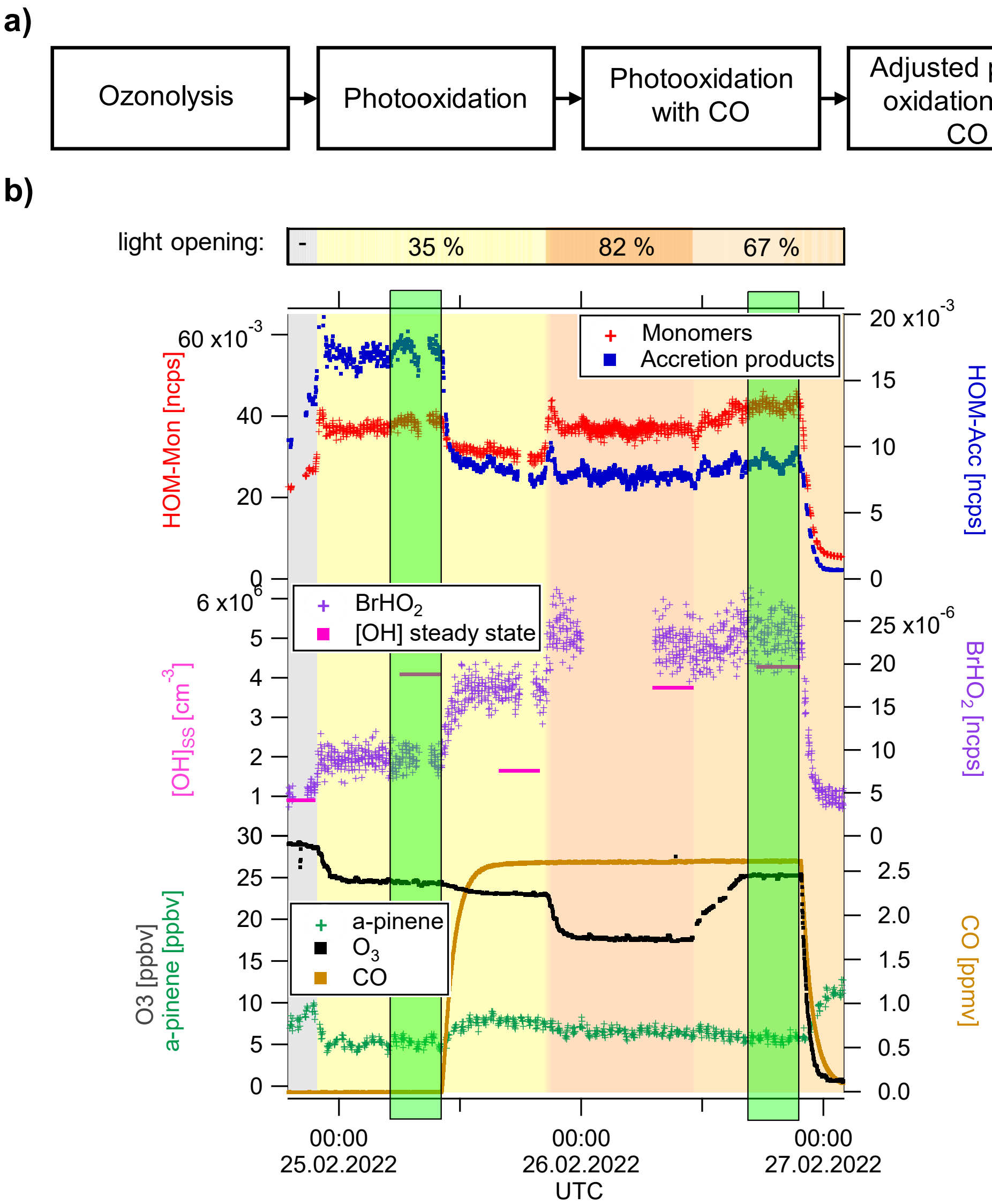

**Figure 2: a) Experiment flow scheme b) Exemplary timeseries of *Exp1* experiment showing HOM-Mon and HOM-Acc product sum (top panel), calculated OH concentration and $BrHO_2$ signal (middle panel), and ozone, α-pinene and CO concentrations (bottom panel). Background color represents light intensity. Highlighted in green are the low $HO_2/RO_2$ steady state and the steady state at high $HO_2/RO_2$ (addition of CO and adjusted oxidant level).**

All experiments started with α-pinene ozonolysis followed by illumination of the UVC-lights to induce the reaction with OH. A general flow scheme of the experiment can be found in **Fig. 2**, together with one exemplary timeseries of the unseeded experiment *Exp1*. After the photooxidation steady state, CO was added to the system. In the displayed *Exp1* experiment the OH concentration was adjusted in three steps to approach the desired value. First the UVC-light opening was adjusted and then $O_3$ was added, and the UVC-light opening was adjusted again. In some experiments initially the effect of

CO on the unchanged system was observed, before the adjustment of OH. In other experiments (*Exp2.2*, *Exp3*) the adjustment of the α-pinene OH turnover via ozone concentration and UVC-light opening were made simultaneously with the CO addition. Highlighted in green are the steady states with the "same" OH concentration characterized by low and high $HO_2/RO_2$, which were used for analysis and interpretation.

## 2.4 Model calculation for $HO_2/RO_2$ ratio estimation

Box model calculations were performed applying the MCM v3.3.1 chemistry (Jenkin et al., 1997; Saunders et al., 2003) under the boundary conditions of the SAPHIR-STAR chamber. All calculations were performed with the institute software package EASY which uses FACSIMILE to solve the differential equations (EASY Version 5.69b). More details about the model parameters can be found in the supplement **Sect. S3**. The model calculations reproduced the primary observables α-pinene, $O_3$, CO, and OH within the experimental uncertainties. The box-model results were used to characterize the $HO_2/RO_2$ ratio of the chemical systems, as no direct measurement of these parameters was available. The observed cluster signal $BrHO_2^-$ follows the modelled $HO_2$ concentration (**Fig. 3**).

The model predicts a shift of the $HO_2/RO_2$ ratio from about 0.01 to about 1 by CO addition and oxidant adjustment, an increase by two orders of magnitude. Owing to lack of observations to verify model results, we will consider only the magnitude of $HO_2/RO_2$ here. The model results show that indeed a major shift from $RO_2$+$RO_2$ to $RO_2$+$HO_2$ reactions can be expected.

We further used the modelled $RO_2$ and $HO_2$ concentrations to estimate the relative importance of pathways for individual (observed) HOM-RO2. For that we applied two generic rate coefficients $k_{RO2HO2}$ and $k_{RO2RO2}$. As the rate coefficient for the $RO_2$+$HO_2$ termination to a hydroperoxide $k_{RO2HO2}$ we used the value specified in the MCM ($1.85 \cdot 10^{-11}$ $cm^3 \cdot s^{-1}$ at 20 °C (Jenkin et al., 1997; Saunders et al., 2003)). We chose a $k_{RO2RO2}$ of $5 \cdot 10^{-12}$ $cm^3 \cdot s^{-1}$ as the approximated reaction rate of the $RO_2$+$RO_2$ reactions. This value applies to all possible reactions (accretion product, monomer, and alkoxy formation) and is in the range of $k_{RO2RO2}$ utilized by Roldin et al. (2019) in the PRAM model.

## 2.5 Determination of oxidized VOCs, HOMs and $HO_2$

Chemical ionization mass spectrometry (HR-TOF-CIMS) techniques were used to detect a range of gaseous compounds. For this, two atmospheric pressure interface time of flight mass spectrometers (APi-TOF-MS; Tofwerk AG) with different inlet systems were used simultaneously. General information about the APi-TOF-MS instrument can be found in Junninen et al. (2010).

A long TOF (LTOF) (Resolution ~8500 for peaks at >200 m/Q) was coupled with the multi-scheme ionization inlet (MION; Karsa Oy). The setup of the inlet is described in detail by Rissanen et al. (2019). The distinctive feature of the MION inlet is the switching between two reagent ions. Here, nitrate was used to detect closed-shell HOMs, as well as HOM-$RO_2$. Bianchi et al. (2019) suggested to define HOM as products stemming from autoxidation containing more than 6 oxygen. In our

overall analysis we decided to also include fragments and monomers containing 5 or in a few cases 4 oxygens (see peaklist in supplement **Sect. S4**) as we are interested to see if the importance of these less oxidized (but still with $NO_3$-CIMS detectable) products increases at higher $HO_2/RO_2$. However, in all considerations regarding SOA formation we furthermore set a molar weight threshold which automatically excluded any products with less 6 oxygens.

As the second reagent ion, bromide was used to detect less oxidized products and the $HO_2$ radical (Albrecht et al., 2019; Sanchez et al., 2016). The nitrate ion source had a reaction time of 600 ms, while the bromide ion source had a shorter reaction time of 60 ms. For all experiments an inlet flow of 10 L min$^{-1}$ was used and the ionization scheme was switched every 10 minutes.

In the data evaluation the first step was the separation of the timeseries of the two reagent ions. The data was subsequently processed with Tofware (Version 3.2.3, Tofwerk AG) using the high resolution timeseries workflow. No transmission correction was performed as previous measurements showed an approximately flat relative transmission curve in the mass region of interest. The analyte signals were normalized with the reagent ion signal ($NO_3^-$ and $HNO_3NO_3^-$ for nitrate and $Br^-$ and $BrH_2O^-$ for bromide).

Since no direct $HO_2$ calibration was available, the $HO_2$ signal in the Br-MION-CIMS was used to compare the levels of $HO_2$ relative to each other in the different phases of the experiment. The comparison of the measured $HO_2$ signal to the modelled $HO_2$ concentration shows a good linear relation between the model predictions and observations.

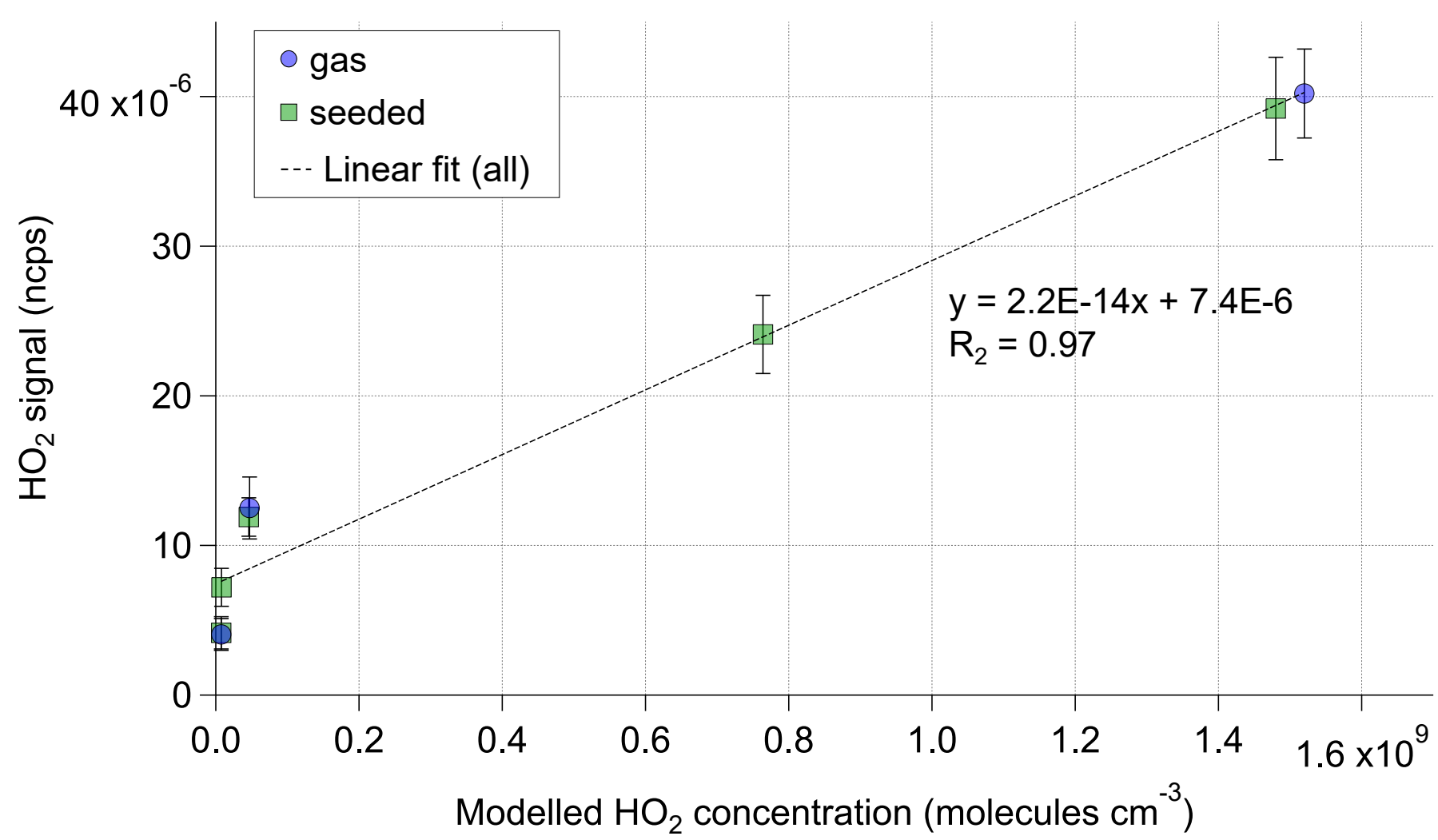

**Figure 3: Modelled $HO_2$ concentration vs. normalized $HO_2$ signal for each steady state of <u>Exp2</u>. $HO_2$ is measured as the $BrHO_2$ cluster and is normalized with the sum of the reagent ion $Br^-$ and its water cluster. The dotted line shows the linear fit to all (gas phase and seeded) measurement points.**

**Figure 3** illustrates this for the example of the *<u>Exp2</u>* experiment. A background signal of around ~$1\cdot10^{-5}$ is observed as soon as VOC and ozone are present in the reactor. The background $HO_2$ signal was not observed when only $O_3$ or only VOC were

in the system. As shown by the MCM modelling results $HO_2$ production of this strength is not expected in the α-pinene ozonolysis phase but this background phenomenon was observed before (Albrecht et al., 2019) and is not fully understood.

For the HOM molecules measured by the $NO_3$-MION-CIMS the relative changes between different experiment phases are compared. For all detected HOM products the same detection sensitivity is assumed. Hyttinen et al. (2018) showed in quantum chemical calculations that HOMs containing 6 or more oxygen atoms have comparable sensitivity with the nitrate reagent ion. At this degree of oxidation it can be expected that the HOMs already contain multiple hydroperoxyl and/or hydroxy functional groups (Bianchi et al., 2019) prior to the termination step, making it unlikely that the sensitivity is strongly influenced by the termination group. Thus, the signal strength reflects the correct ranking of the observations and relative comparisons do not require calibration. Pullinen et al. (2020) studied the mass balance between condensable HOMs and formed particle mass and were able to find closure within a factor of 2.

A second CI-APi-TOF was used to measure less oxidized species. It was configured with a CI inlet based on the design of Eisele and Tanner (1993) coupled to an HTOF (Resolution ~2700 for peaks at >200 m/Q) (Tofwerk AG) and was operated in positive mode with propylamine ($C_3H_7NH_2$, Sigma-Aldrich, purity ≥99%) to detect the early generation $RO_2$ and oxidation products (Berndt et al., 2018). The propylamine was purified and added as an amine-$N_2$ mixture (flow: 0.12 mL $min^{-1}$) to the 30 L $min^{-1}$ sheath flow. Furthermore, the sheath flow air is humidified to optimize ionization. The instrument sampled 0.1 L $min^{-1}$ from the chamber, which was diluted with 9.9 L $min^{-1}$ for a sample flow of 10 L $min^{-1}$. The dilution was necessary to reduce depletion of the primary ion (Hantschke, 2022).

## 3 Results and Discussion

In order to understand the effect of $HO_2/RO_2$ on the gas phase product composition, we will present and compare two cases: The steady state without CO (low $HO_2/RO_2$) and the steady state with CO addition and OH adjustment by $J(O^1D)$ and $O_3$ (high $HO_2/RO_2$). The modelling results predicted $HO_2/RO_2$ of about 1/100 and of about 1/1 for these two cases respectively. The modelling results show that the $HO_2/RO_2$ ratio changes by two orders of magnitude, because [$RO_2$] was reduced by about a factor of three, while [$HO_2$] was increased by a factor of 30. Consequently, $HO_2$ reactions were almost negligible at low $HO_2/RO_2$ while $RO_2$+$RO_2$ reactions can still contribute at high $HO_2/RO_2$.

Assuming correctly modelled [$HO_2$] and [$RO_2$], we calculated the competition between $HO_2$ and $RO_2$ reactions for each (observed) $RO_2$ expressed in form of pseudo first order rate coefficients in $k_{RO_2HO_2} \cdot [HO_2]$ or $k_{RO_2RO_2} \cdot [RO_2]$. Herein [$RO_2$] is the sum of all $RO_2$ species as defined in the MCM v3.3.1. For all experiments the results of our calculations indicate that the sink for HOM-$RO_2$ is dominated by $RO_2$+$RO_2$ reactions at low $HO_2/RO_2$ (~98 % contribution), while at high $HO_2/RO_2$ $RO_2$+$HO_2$ contributed ~75 %. As the rate coefficients are not well known and we cannot verify the modelling results for $HO_2$ and $RO_2$ our calculations serve solely as an indication of expected trends in the chemical system.

### 3.1 Impact on overall HOM-formation

The top panel of **Fig. 2** shows the timeseries of HOM-Mon and HOM-Acc products. The HOM-Mon signal recovers after the oxidant adjustment, while the HOM-Acc signal is significantly suppressed at high $HO_2/RO_2$. This indicates that the shift from low to high $HO_2/RO_2$ substantially impacts the termination reactions, shifting formation from the HOM-Acc product channel ($RO_2$+$RO_2$) to the HOM-Mon channel.

An overview of the results for the product classes defined in the method section is shown in **Fig. 4**. Plotted are the average ratios of signal in the $NO_3$-CIMS in the high $HO_2/RO_2$ steady state compared to the low $HO_2/RO_2$ steady state. For better comparison, all experiment phases were normalized to the actual α-pinene OH turnover. The overall HOM-signal was lower at high $HO_2/RO_2$ showing a reduction of about 20 %. Most distinctive, the HOM-Acc were strongly reduced by about 60 %. A reduction of HOM-Acc by addition of CO was observed before by McFiggans et al. (2019), however there the OH concentration was not kept constant. The HOM-Frag ($5 \geq C < 10$) also show a reduction of about 20 %. At high $HO_2/RO_2$ $C_{10}$-HOM-$RO_2$ were also reduced significantly by about 40 %.

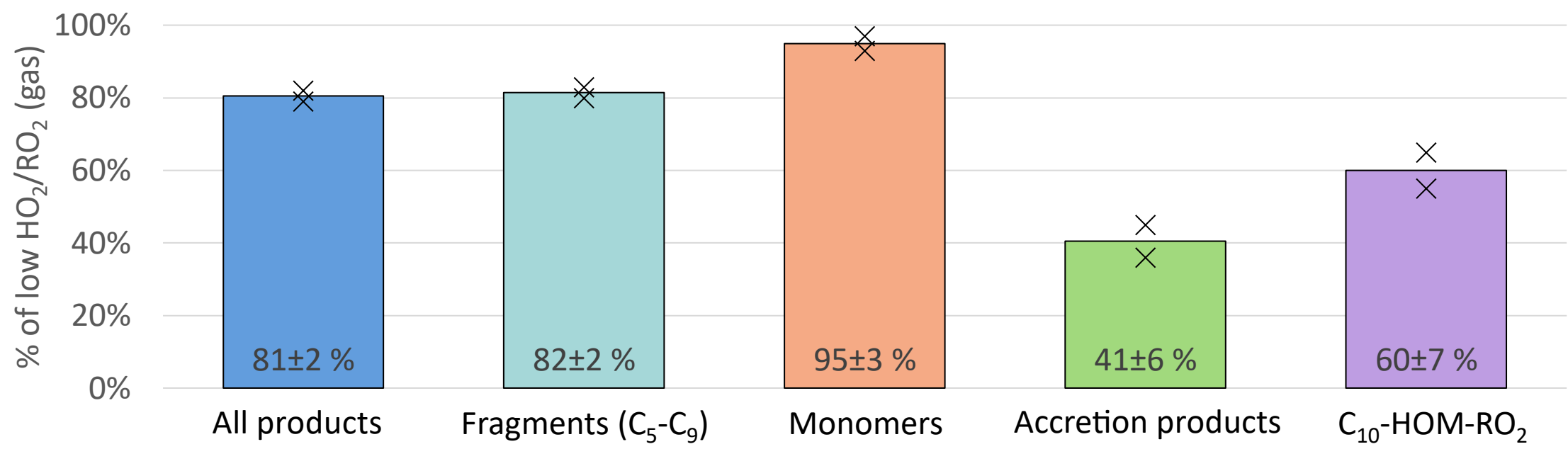

**Figure 4: Overview of average, relative change in product classes detected in $NO_3$-CIMS between low and high $HO_2/RO_2$ case (both normalized to α-pinene OH turnover) for pure gas phase experiments. Bars represent average of the two experiments, markers represent individual experiments.**

The HOM-Mon signal level remained about the same at low and high $HO_2/RO_2$. Without reduction in the HOM-$RO_2$ precursors a reduction of HOM-Acc should lead to an increase in HOM-Mon, as each HOM-Acc is formed from one HOM-$RO_2$ (HOM-$RO_2$+$RO_2$) or potentially even two HOM-$RO_2$ (HOM-$RO_2$+HOM-$RO_2$). Of course, the presence of $HO_2$ could reduce the alkoxy formation, and thus fragmentation of HOM-$RO_2$. This missing sink could lead to an additional HOM-Mon source compared to the low $HO_2/RO_2$ case. However, the distribution of the product classes at low and high $HO_2/RO_2$ (**Fig. 5**) shows that contributions are shifted from HOM-Acc to HOM-Mon, while the contribution of HOM-Frag remains constant.

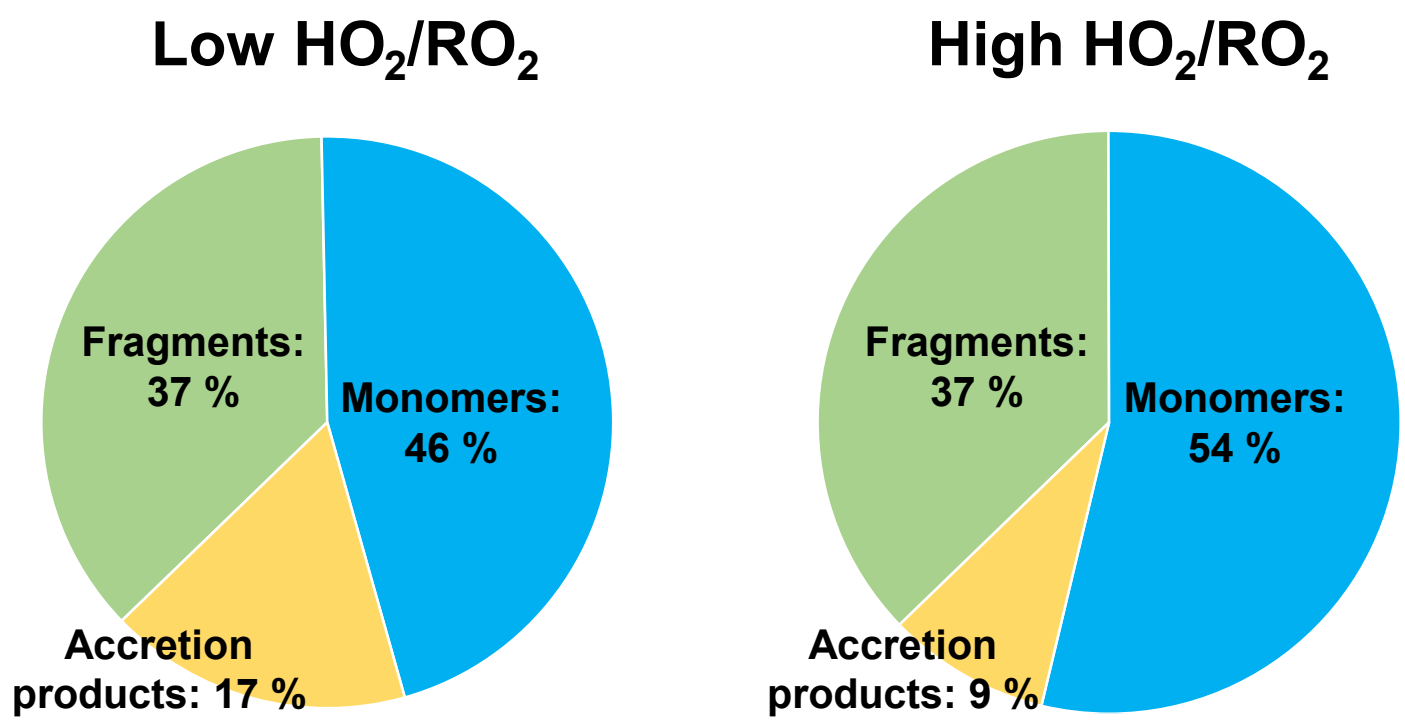

**Figure 5: Average contribution of the closed shell product classes to overall HOM-product signal in the low and high $HO_2/RO_2$ cases (pure gas phase experiments).**

Further changes in the product distribution become evident when considering the individual HOM-Mon families as shown in **Fig. 6**. The $C_{10}H_{15}O_X$ peroxy radical family and the related $C_{10}H_{14}O_z$ family (carbonyl compounds) show the strongest suppression with a decrease of about 40 % at high $HO_2/RO_2$. For the $C_{10}H_{17}O_X$ peroxy radical family the suppression was less pronounced with a 17 % reduction. In contrast, the $C_{10}H_{16}O_z$ family remained about the same while the $C_{10}H_{18}O_z$ family showed a strong increase at high $HO_2/RO_2$.

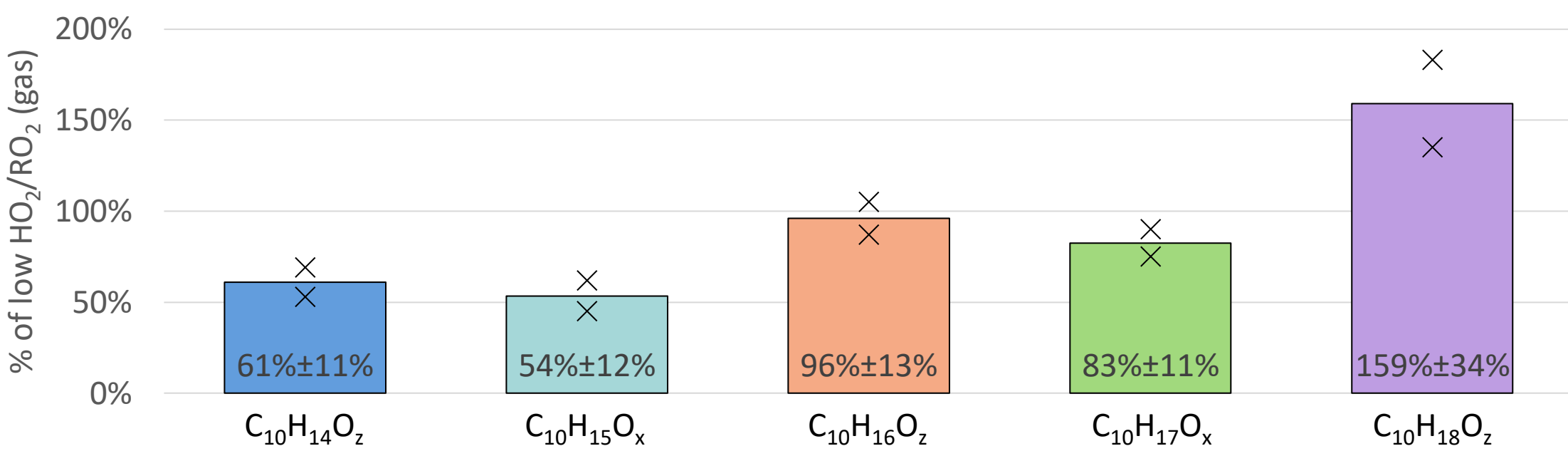

**Figure 6: Overview of average, relative change in monomer families detected in $NO_3$-CIMS between low and high $HO_2/RO_2$ case (both normalized to α-pinene OH turnover) for pure gas phase experiments. Bars represent average of the two experiments, markers represent individual experiments.**

The suppression of $C_{10}$-HOM-$RO_2$ of only about 40 % compared to the reduction of overall [$RO_2$] by ~70 % in the model calculations shows that in many instances the autoxidation is too efficient to be out-competed by the $RO_2$+$HO_2$ termination reaction, which is several times faster than $RO_2$+$RO_2$ reactions.

Furthermore, the signal weighted O/C ratio of the monomer class does not change between low and high $HO_2/RO_2$ (0.70± 0.01). If the $HO_2$ termination would interrupt the autoxidation chain, a lower oxidation level would be expected at high $HO_2/RO_2$. The unchanged oxidation level and the suppression of HOM-Acc, indicate that the average autoxidation rate must be faster than $k_{RO2HO2}$·[$HO_2$], while the average accretion rate for $k_{HOM\text{-}RO2+RO2}$·[$RO_2$] must be slower. In conclusion, the change in $HO_2/RO_2$ should essentially impact the distribution of the HOM-$RO_2$ termination products.

### 3.2 Impact on HOM-$RO_2$

$C_{10}$-HOM-$RO_2$ are key to understand the changes in the HOM product distribution. Therefore, we will first discuss the changes in the HOM-$RO_2$ and then the changes in the closed shell products.

The $C_{10}$ peroxy radical class consists of the $C_{10}H_{15}O_x$ and $C_{10}H_{17}O_x$ families which were reduced to 54 % and 83 %, respectively when comparing the high and low $HO_2/RO_2$ cases (**Fig. 6**, light blue and green bars). The observed reduction in $C_{10}$-HOM-$RO_2$ is significantly smaller than the overall $RO_2$ concentration reduction predicted by the MCM model results (reduction to ~30 %). In the following paragraphs, we present a plausibility consideration to assess if these observed changes are consistent with our expectations from modelling results and reaction rates.

The change in the steady state concentration of a compound is always defined by the changes in its sources and sinks. The source of a HOM-$RO_2$ is the intramolecular reaction of a precursor $RO_2$ and thus the HOM-$RO_2$'s source is reduced if the steady state concentration of the precursor $RO_2$ is reduced. However, assuming the source term of the precursor $RO_2$ is the same in low and high $HO_2/RO_2$ (due to the constant α-pinene OH turnover) and the precursor $RO_2$'s sink term is dominated by the fast autoxidation in both cases, then the $RO_2$'s steady state concentration would not be significantly changed. This consideration is only applicable for $RO_2$ where autoxidation dominates the sink term at low and high $HO_2/RO_2$. However, the unchanged oxidation level of the HOM-Mon indicates that once the autoxidation is initiated it out-competes the possible termination reactions.

In this case, the change in steady state concentration of the HOM-$RO_2$ will be defined by the changes in the sink terms. Owing to the faster reaction of $RO_2$+$HO_2$ compared to $RO_2$+$RO_2$ the chemical sink for all $RO_2$ including HOM-$RO_2$ with slower autoxidation rates increased, which leads to a reduction in the steady state concentration of $RO_2$ in general, despite holding the primary $RO_2$ source term constant.

For steady state conditions, we can estimate the expected effect of high and low $HO_2/RO_2$ on the $RO_2$ ratio for those HOM-$RO_2$ with production directly linked to the primary production ($k_{OH}\cdot[OH]\cdot[\alpha\text{-}pinene]$) with negligible further autoxidation. The necessary equations and assumptions can be found in supplement **Sect. S5.** We assume the same primary production at low and high $HO_2/RO_2$ and that the reaction with $HO_2$, the reaction with $RO_2$ and the wall loss are the only significant loss pathways. At high $HO_2/RO_2$, a reduction to 80 % is expected if the rate coefficient $k_{RO2HO2}$ ($1.85\cdot10^{-11}$ $cm^3\cdot s^{-1}$ at 20 °C (Jenkin et al., 1997; Saunders et al., 2003)) is 5 times faster than $k_{RO2RO2}$ (leading to $k_{RO2RO2}=3.7\cdot10^{-12}$ $cm^3\cdot s^{-1}$). A reduction to 60 % is expected if $k_{RO2HO2}$ is 8 times faster than $k_{RO2RO2}$. These reductions are in the range of what is observed for the $C_{10}$-HOM-$RO_2$. Of course, the approach of using generalized bulk rate constants is limited, but the resulting values for $k_{RO2RO2}$ were clearly within the range of rate coefficients expected for HOM-$RO_2$+$RO_2$ reactions (Roldin et al., 2019) showing that the increased chemical sink is a plausible explanation for our observations.

The $C_{10}H_{15}O_x$ family is on average reduced by around 30 % more than the $C_{10}H_{17}O_x$ family (see **Fig. 6**). $C_{10}H_{15}O_X$ peroxy radicals are either formed by sequential oxidation of α-pinene , e.g. from oxidation products like pinonaldehyde, or directly

from α-pinene via the H-abstraction pathway (Shen et al., 2022). Formation of pinonaldehyde and, even more so HOM formation via the H-abstraction channel, involve alkoxy steps. However, alkoxy radicals should be reduced at high $HO_2/RO_2$ since they are mainly formed by $RO_2+RO_2$ reactions in the absence of $NO_x$. Thus, missing source terms add to the increased chemical sink by $HO_2$ for $C_{10}H_{15}O_X$ peroxy radicals.

Amine CIMS measurements enabled detection of the formula composition $C_{10}H_{16}O_2$ (e.g. pinonaldehyde). $C_{10}H_{16}O_2$ was reduced on average to 71%±1 % at high $HO_2/RO_2$ compared to low $HO_2/RO_2$. This supports that a fraction of the $C_{10}H_{15}O_x$ radical decrease at high $HO_2/RO_2$ arose from suppression of $C_{10}H_{16}O_2$ first generation products. In addition, a further suppression of HOM formation via the H-abstraction channel is likely. It should be noted that the reduction of $C_{10}H_{16}O_2$ is smaller than that expected by the MCM model results. This might indicate that $HO_2$ can also enable alkoxy radical steps to a certain degree as summarized by Jenkin et al. (2019) and postulated by e.g. Eddingsaas et al. (2012) as a source of pinonaldehyde in $HO_2$ dominated systems.

According to the model calculations the pseudo first order rate coefficient $k_{RO2HO2}\cdot[HO_2]$ is expected to be about 0.03 $s^{-1}$ for the $RO_2+HO_2$ reaction at high $HO_2/RO_2$. Consequently, only such HOM-$RO_2$ with autoxidation rates of ≤0.03 $s^{-1}$ will be significantly lost by reaction with $HO_2$ at the higher $HO_2$ concentrations. However, typical isomerization rates of peroxy radicals in autoxidation are of the order of 0.1 $s^{-1}$ and many are faster (Piletic and Kleindienst, 2022; Berndt, 2021). Therefore, reduction in a HOM-$RO_2$ is only expected when the faster termination rate of $k_{RO2HO2}\cdot[HO_2]$ can compete with the autoxidation rate, i. e. when the autoxidation slows as the degree of oxidation increases on the specific HOM-$RO_2$.

The increase in chemical sink strength by going from $RO_2$ termination to $HO_2$ termination is the main expected reason for the decrease in $C_{10}H_{17}O_x$. As discussed, the $C_{10}H_{15}O_X$ family is subject to an additional decrease in the precursors due to the alkoxy steps necessary in the formation pathway. Since $C_{10}H_{15}O_X$ were the main contributors to the $C_{10}$-HOM-$RO_2$ class their stronger reduction is reflected in the overall reduction of $C_{10}$-HOM-$RO_2$.

### 3.2.1 Contribution of $C_{10}H_{15}O_x$ and $C_{10}H_{17}O_x$ families to HOM-$RO_2$

In the pure gas phase experiments, the contribution of the $C_{10}H_{17}O_x$ family to the $C_{10}$-HOM-$RO_2$ class is 23 % ±2 % on average in the low $HO_2/RO_2$ case. In the high $HO_2/RO_2$ case the contribution increases to 31 % ±4 % on average. As discussed above the suggested pathways to $C_{10}H_{15}O_x$ HOM-$RO_2$ may be additionally suppressed due to a decrease of alkoxy steps at high $HO_2/RO_2$ reducing the entry channel into $C_{10}H_{15}O_x$ HOM-$RO_2$.

Nevertheless, the contribution of $C_{10}H_{15}O_x$ is substantial in both experiment stages. Kang (2021) and Shen et al. (2022) reported that, in the photooxidation of α-pinene, the HOM-$RO_2$ detected by $NO_3$-CIMS are dominated by the $C_{10}H_{15}O_x$ family, while $C_{10}H_{17}O_x$ formation is the main expected OH reaction pathway described in literature (Berndt, 2021; Berndt et al., 2016; Xu et al., 2019).

This hints towards an effective pathway to HOM via $C_{10}H_{15}O_x$. A reason may be the fast opening of both carbon-rings in the bicyclic α-pinene (Shen et al., 2022), or a four-ring opening in pinonaldehyde or similar compounds, for easy autoxidation. From our observations increasing the $HO_2/RO_2$ ratio does increase the relative importance of the $C_{10}H_{17}O_X$ family, but the change is less than 10 % in contribution.

Contribution of the two peroxy radical families to the HOM formation is also reflected in the composition of $C_{20}$ HOM-Acc. **Figure 7** shows the average contributions of the $C_{20}H_{30}O_z$, $C_{20}H_{32}O_z$, and $C_{20}H_{34}O_z$ families in the low and high $HO_2/RO_2$ cases. Although the absolute amount of HOM-Acc was suppressed by 60 % the family distribution was similar, $C_{20}H_{32}O_z$ dominated, while $C_{20}H_{30}O_z$ was lowest. $C_{20}H_{30}O_z$ is formed from two members of the $C_{10}H_{15}O_x$ family, while $C_{20}H_{34}O_z$ is formed by two members of the $C_{10}H_{17}O_x$ family. $C_{20}H_{32}O_z$ is then a combination of a $C_{10}H_{15}O_x$-$RO_2$ and $C_{10}H_{17}O_x$-$RO_2$.

Families that require one or two $C_{10}H_{17}O_X$ peroxy radicals for their formation have a higher contribution than the $C_{10}H_{17}O_x$ family's contribution to $C_{10}$-HOM-$RO_2$. Here, it is important to note that not only HOM-$RO_2$ can participate in HOM-Acc formation, but also traditional, less oxidized $RO_2$ radicals (Berndt et al., 2018; Pullinen et al., 2020; McFiggans et al., 2019), which are not detectable by $NO_3^-$-CIMS. However, more oxidized peroxy radicals exhibit faster accretion rates (Berndt et al., 2018).

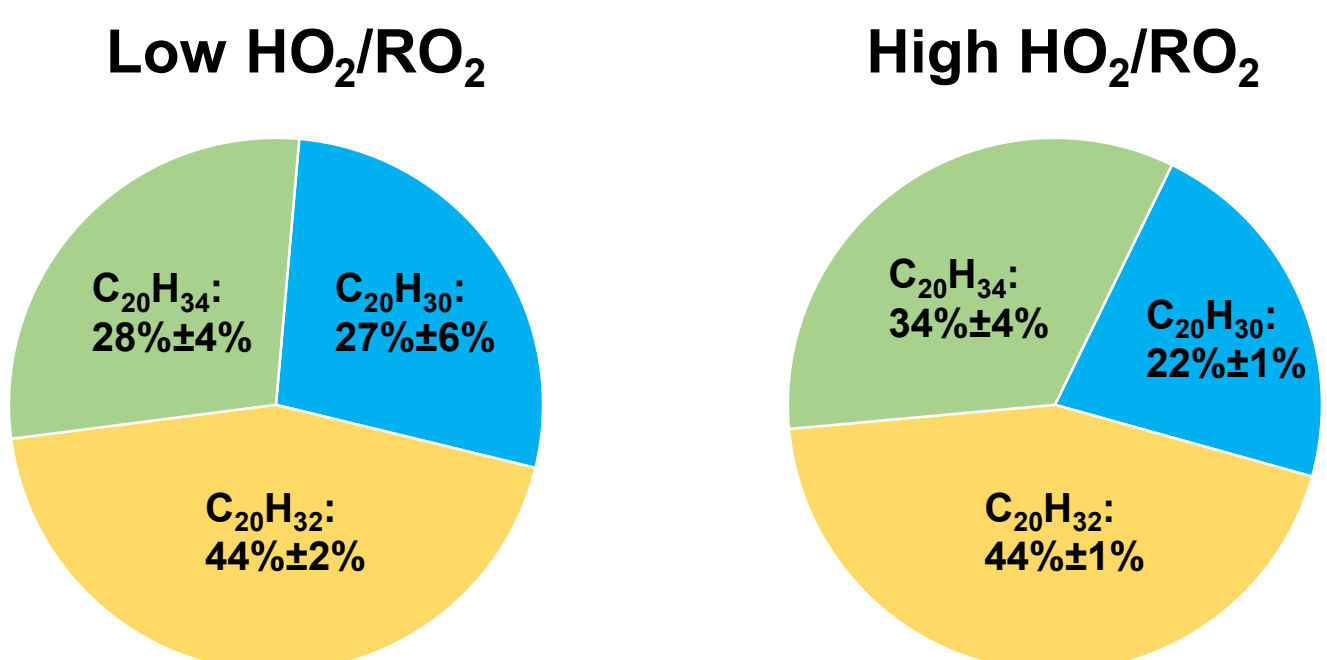

**Figure 7: Average contribution of the $C_{20}H_{30}O_z$, $C_{20}H_{32}O_z$, and $C_{20}H_{34}O_z$ family to the $C_{20}$ HOM-Acc group signal in the low and high $HO_2/RO_2$ cases (pure gas phase experiments). Not pictured is $C_{20}H_{28}O_z$ due to its negligible signal (contribution ~1 %).**

The large contributions of $C_{20}H_{32}O_z$ and $C_{20}H_{34}O_z$ thus clearly show the general importance of the $C_{10}H_{17}O_X$ peroxy radicals. The largest fraction, the $C_{20}H_{32}O_z$ family reflects the importance of HOM-$C_{10}H_{15}O_x$ and the high abundance of lower oxidized $C_{10}H_{17}O_X$ peroxy radicals. The fraction of $C_{20}H_{34}O_z$ is smaller because their formation requires HOM-$C_{10}H_{17}O_X$ radicals which are less abundant compared to HOM-$C_{10}H_{15}O_x$, while the small fraction of $C_{20}H_{30}O_z$ indicates that, despite the importance of HOM-$C_{10}H_{15}O_x$, lower oxidized $C_{10}H_{15}O_x$ are less important.

These results indicate the importance of mixed HOM-Acc formation by cross reactions of HOM-$RO_2$ and a lower oxidized $RO_2$. The importance of mixed HOM-Acc is supported by the relatively small fractions of HOM-Acc products with very high oxygen numbers, which more likely stem from HOM-$RO_2$+HOM-$RO_2$. For example, $C_{20}$-HOM-Acc with 12 or more

oxygen atoms contribute only around 30 % (low $HO_2/RO_2$: 26 % ±4 %, high $HO_2/RO_2$: 31 % ±2 %) of the signal in the product group.

Although the effect of the changed $HO_2/RO_2$ ratio is small, a tendency to higher $C_{20}H_{34}O_z$ contribution was observed. This is consistent with the observation of a slightly higher $C_{10}H_{17}O_x$ contribution to the $C_{10}$-HOM $RO_2$. The stronger suppression of the $C_{10}H_{15}O_x$ family at high $HO_2/RO_2$ is the first indication for, and can be explained by, a reduction in the alkoxy radical formation.

### 3.2.2 Impact on HOM-Alkoxy radical formation

Alkoxy radicals (RO) are the second important radical type in the oxidation chain of α-pinene. RO cannot be detected directly as they are highly unstable and thus have very low concentrations. However, as explained in **Sect.** Fehler! Verweisquelle konnte nicht gefunden werden. the parity change in the HOM-$RO_2$ families can be used as a diagnosis tool for the abundance of alkoxy steps (Kang, 2021). A second indicator for alkoxy steps is the abundance of HOM products with less than 10 C-atoms.

**Figure 8** shows the average contribution of $C_{10}H_{15}O_x$ and $C_{10}H_{17}O_x$ with an even and odd number of oxygens at low and high $HO_2/RO_2$. $C_{10}H_{15}O_x$ radicals with an even number of oxygens contribute on average 32 % at low $HO_2/RO_2$. For $C_{10}H_{15}O_x$, the autoxidation chain is expected to start from an even number of oxygen either from $C_{10}H_{15}O_4$ (pinonaldehyde-like) (MCM v3.3.1 (Jenkin et al., 1997; Saunders et al., 2003) or from $C_{10}H_{15}O_2$ ($C_{10}H_{16}$ H-abstraction) (Berndt, 2021; Shen et al., 2022). Therefore, without the involvement of an alkoxy step, the parity of the oxygen number in the observed $C_{10}H_{15}O_x$ HOM-$RO_2$ is expected to be even. Due to the average contribution of $C_{10}H_{15}O_{odd}$ of 69 % we conclude that at least one alkoxy step (or any odd number of alkoxy steps) must have taken place in most of the cases at low $HO_2/RO_2$.

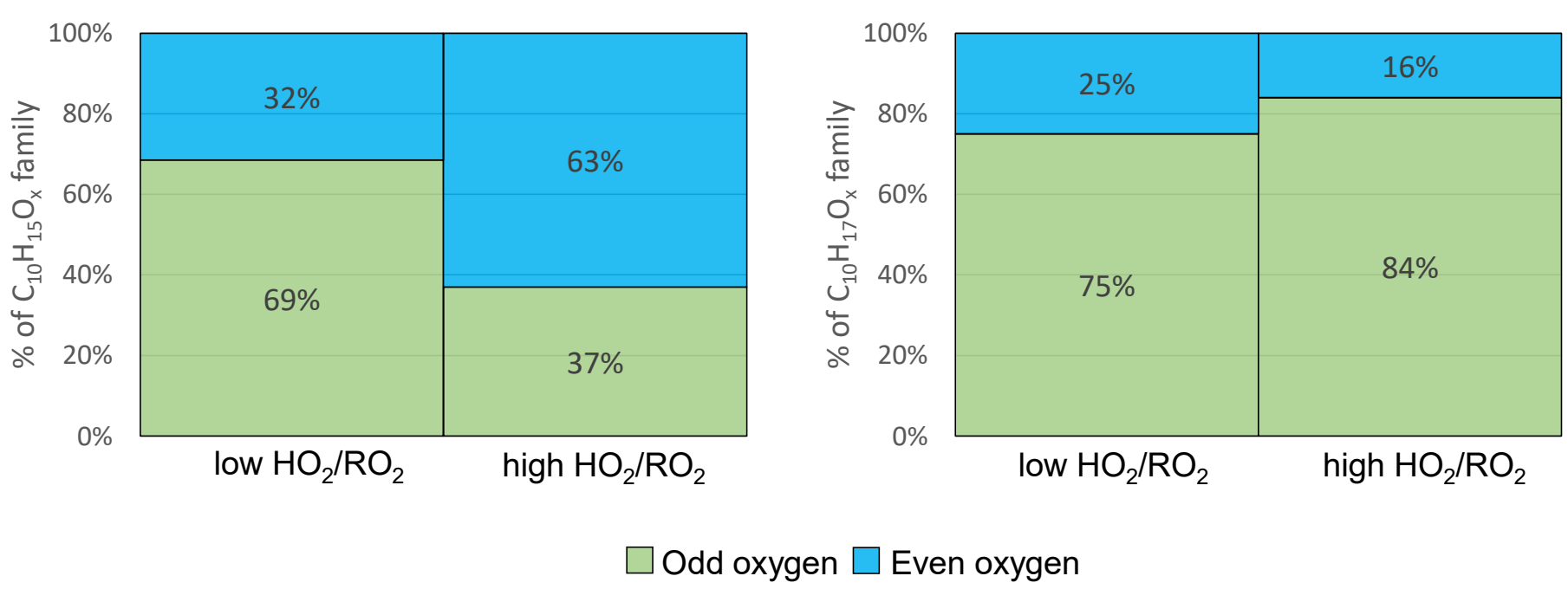

**Figure 8: Average contribution of $O_{odd}$ and $O_{even}$ to the HOM-$RO_2$ families $C_{10}H_{15}O_x$ (left) and $C_{10}H_{17}O_x$ (right) signal in the low and high $HO_2/RO_2$ cases (pure gas phase experiments).**

At high $HO_2/RO_2$ $C_{10}H_{15}O_{even}$ contributed 63 % and the $C_{10}H_{15}O_{odd}$ contribution was reduced to 37 %. This demonstrates a change in the number of alkoxy steps along the formation pathway of the observed HOM-$RO_2$ radicals. The increased contribution of $C_{10}H_{15}O_{even}$ at high $HO_2/RO_2$ lets us infer an even number of alkoxy steps as more common (0,2,4…). In the

simplest case 1 or 2 alkoxy step take place at low $HO_2/RO_2$ due to HOM-RO formation from HOM-$RO_2$+$RO_2$ reactions, while no or 1 alkoxy step take place at high $HO_2/RO_2$, because HOM-$RO_2$+$HO_2$ produces none or less HOM-RO than HOM-$RO_2$+$RO_2$.

For $C_{10}H_{17}O_x$ the entry channel into autoxidation is $C_{10}H_{17}O_3$ with an odd number of oxygen atoms. Therefore, in autoxidation without alkoxy steps the oxygen parity is expected to be odd. At low $HO_2/RO_2$ $C_{10}H_{17}O_{odd}$ species contribute 75 % to the total $C_{10}H_{17}O_x$ signal indicating that either none or an even number (2,4,…) of alkoxy steps occurred. At high $HO_2/RO_2$ the odd contribution increases to 84 % (see **Fig. 8**). This result could indicate a low occurrence of alkoxy steps even at low $HO_2/RO_2$, with a further decrease of alkoxy formation at high $HO_2/RO_2$. However, the observed shift is minor.

In any case the different responses of the $C_{10}H_{15}O_x$ and $C_{10}H_{17}O_x$ families to the reduction of HOM-$RO_2$ formation from HOM-$RO_2$+$RO_2$ at high $HO_2/RO_2$ indicate that there could be fundamental differences in the autoxidation chains of $C_{10}H_{15}O_x$ and $C_{10}H_{17}O_x$ (or the limit of the parity analysis). The parity analysis indicates a decrease in alkoxy steps at high $HO_2/RO_2$, but it cannot be directly inferred with certainty. However, decrease in alkoxy steps at high $HO_2/RO_2$ is supported by the observation of changes in HOM-Frag products.

On average the sum of all HOM-Frag products (detected compounds with $5 \geq C < 10$ by $NO_3^-$-CIMS) showed a reduction of around 20 % (pure gas phase experiments, see **Fig. 4**). Further trends become recognizable when separating the species according to their carbon number. **Figure 9** shows the $C_5$, $C_7$, $C_8$, and $C_9$ HOM-Frag at high $HO_2/RO_2$ compared to the low $HO_2/RO_2$ case, normalized to the α-pinene OH turnover. The fragment group with $C_6$ compounds is not included, as it contributed less than 5 % of the fragment signal and contained few detected compounds.

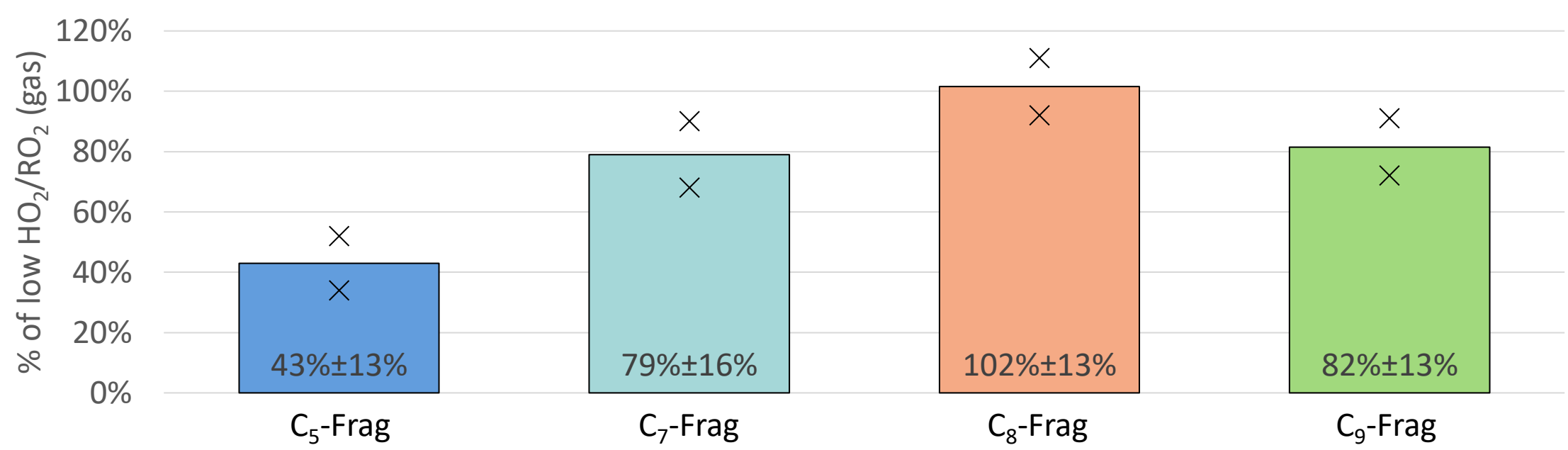

**Figure 9: Overview of average, relative change in $C_5$, $C_7$, $C_8$, $C_9$ fragment groups detected in $NO_3$-CIMS between high and low $HO_2/RO_2$ case (both normalized to α-pinene OH turnover) for pure gas phase experiments. Bars represent average of the two experiments, markers represent individual experiments.**

**Figure 9** shows a significant reduction in HOM-Frag with shorter carbon chain length: $C_5$ HOM-Frag are reduced by around 60 % compared to the low $HO_2/RO_2$ case. If we assume that the fragmentation of $C_{10}$ compounds happens in consecutive steps via scission of HOM-RO radicals (analogously to the MCM), this observation is in accord with decreasing importance of alkoxy radical formation at high $HO_2/RO_2$.

Overall, all observations indicate strong involvement of RO in HOM formation as well as a reduced, but still significant, involvement of RO at high $HO_2/RO_2$, when $HO_2$ chemistry dominates: This is supported by the change of the oxygen parity in $C_{10}$-HOM-$RO_2$, and the decrease of fragmentation products, especially with lower carbon number, as well as the only moderate reduction in the observed $C_{10}H_{16}O_2$ product (pinonaldehyde) and the still substantial importance of the $C_{10}H_{15}O_x$ HOM-$RO_2$ family at high $HO_2/RO_2$.

### 3.3 Impact on carbonyl and hydroperoxide formation

Increased $HO_2/RO_2$ should shift the product distribution by reduction of alcohol and carbonyl compounds from the so-called molecular channel in the $RO_2$+$RO_2$ reaction (see reaction ***(R3)***), in favor of hydroperoxide formation from $RO_2$+$HO_2$ termination (reaction ***(R1)***). This effect can be best observed in the $C_{10}H_{18}O_z$ family, which contains the hydroperoxide and alcohol termination products arising from $C_{10}H_{17}O_x$. $C_{10}H_{18}O_z$ significantly increased to on average 159 % (see **Fig. 6**). This supports an increased hydroperoxide formation, however, with some uncertainty due to the alcohol termination products from $C_{10}H_{17}O_x$ (by reaction with $RO_2$). To elucidate this further the contribution of individual species to the $C_{10}H_{18}O_z$ family was examined.

Formation of an alcohol via the molecular path (reaction ***(R3)***) leads to the loss of one oxygen atom compared to the precursor $C_{10}H_{17}O_x$ radical, while in the hydroperoxide formation (reaction ***(R1)***) the oxygen number remains the same. The most abundant member of the $C_{10}H_{17}O_x$ family was $C_{10}H_{17}O_7$ with a contribution of 72 % ±6 % at low $HO_2/RO_2$, and a contribution of 82 % ±1 % at high $HO_2/RO_2$. $C_{10}H_{17}O_7$ terminates to $C_{10}H_{18}O_z$ products either as an alcohol with sum formula $C_{10}H_{18}O_6$, or as a hydroperoxide with sum formula $C_{10}H_{18}O_7$. These products have additional sources from $C_{10}H_{17}O_6$ and $C_{10}H_{17}O_8$ but due to the dominant contribution of $C_{10}H_{17}O_7$ to the $C_{10}H_{17}O_x$ family we expect any other production channels to be of minor importance.

**Figure 10** shows the HOM product distribution within the $C_{10}H_{18}O_z$ family at low and high $HO_2/RO_2$. The sum of the $O_6$ and $O_7$ product did not change significantly in the two regimes (about 88 %), showing that these are the major products, and agreeing well with the observation of $C_{10}H_{17}O_7$ being the major $C_{10}H_{17}O_x$ HOM-$RO_2$. At low $HO_2/RO_2$ the $O_6$ product has a larger contribution of 64 %±8 %, while at high $HO_2/RO_2$ ~30% of signal is shifted to the $O_7$ product. This shows that the increase in the $C_{10}H_{18}O_z$ is matched with an increase of hydroperoxide formation.

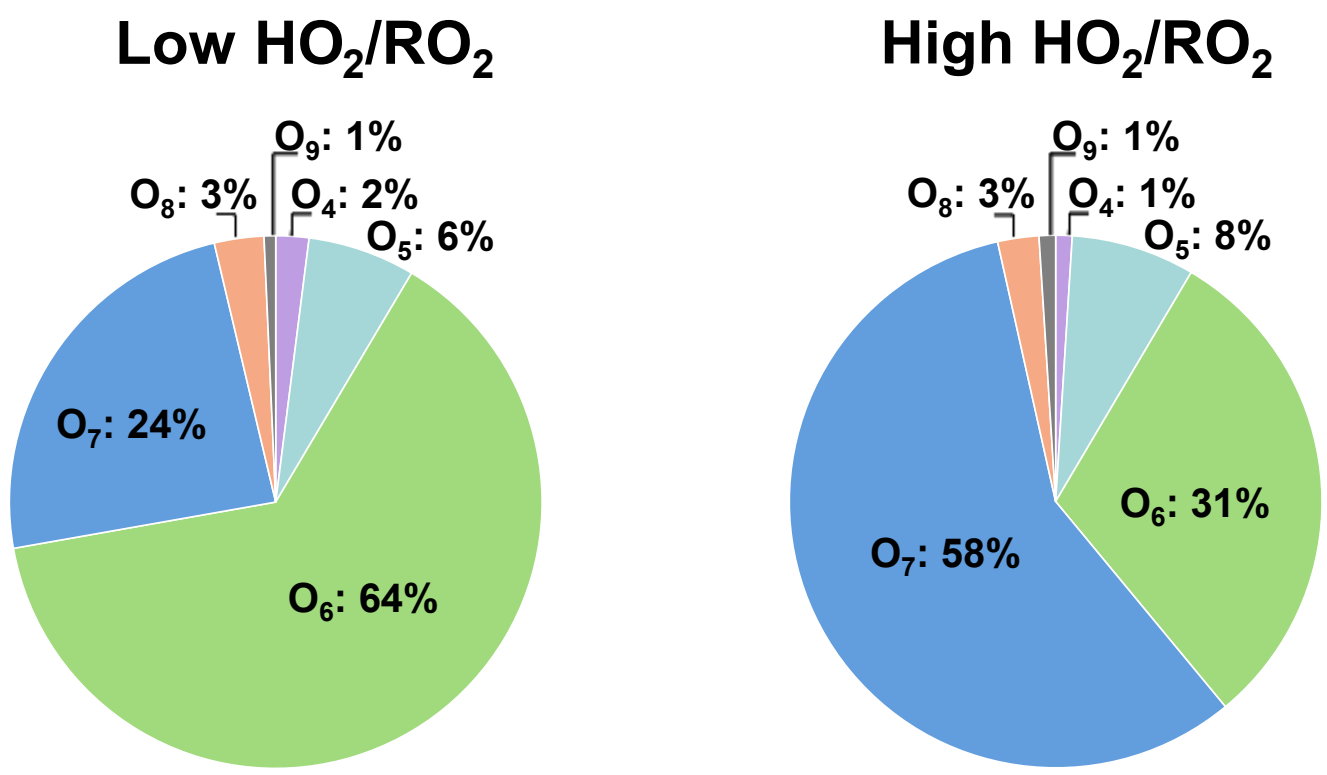

**Figure 10: Average contribution of the individual compounds to the $C_{10}H_{18}O_z$ family signal at low and high $HO_2/RO_2$ (pure gas phase experiments).**

An indicator for carbonyl formation is the $C_{10}H_{14}O_z$ family as it only contains the carbonyl products arising from $C_{10}H_{15}O_x$-$RO_2$. The $C_{10}H_{14}O_z$ family was reduced on average to 61 % at high $HO_2/RO_2$, however this decrease matches the decrease in the $C_{10}H_{15}O_x$ precursor family. If the reaction of a $C_{10}H_{15}O_x$-HOM-$RO_2$ with a second $RO_2$ were the main formation pathway for $C_{10}H_{14}O_z$ a stronger reduction would be expected as both precursor species were decreased significantly. Instead, it appears that $C_{10}H_{14}O_z$ is mainly impacted by the decrease in $C_{10}H_{15}O_x$ as their reductions are similar. A possible explanation could be that intramolecular termination is a major reaction pathway for $C_{10}H_{15}O_x$-$RO_2$, forming $C_{10}H_{14}O_x$-carbonyls. Intramolecular termination of the autoxidation chain has been discussed in the literature for different VOCs (Shen et al., 2021; Guo et al., 2022), Rissanen et al. (2014) discussed the possible importance of the unimolecular termination via an H-shift, followed by formation of a carbonyl functional group and OH loss in the autoxidation chain of cyclohexene. Piletic and Kleindienst (2022) calculated fast reaction rate constants in the range of 1-30 $s^{-1}$ for such intramolecular termination reactions to carbonyls for some $C_{10}H_{17}O_5$ in the α-pinene photooxidation, indicating that this pathway could also be significant for $C_{10}H_{15}O_x$. However, more investigation is necessary.

The overall contributions of the $C_{10}H_{14}O_z$, $C_{10}H_{16}O_z$, and $C_{10}H_{18}O_z$ families to the HOM-Mon class at high $HO_2/RO_2$ are shifted as shown in **Fig. 11**.

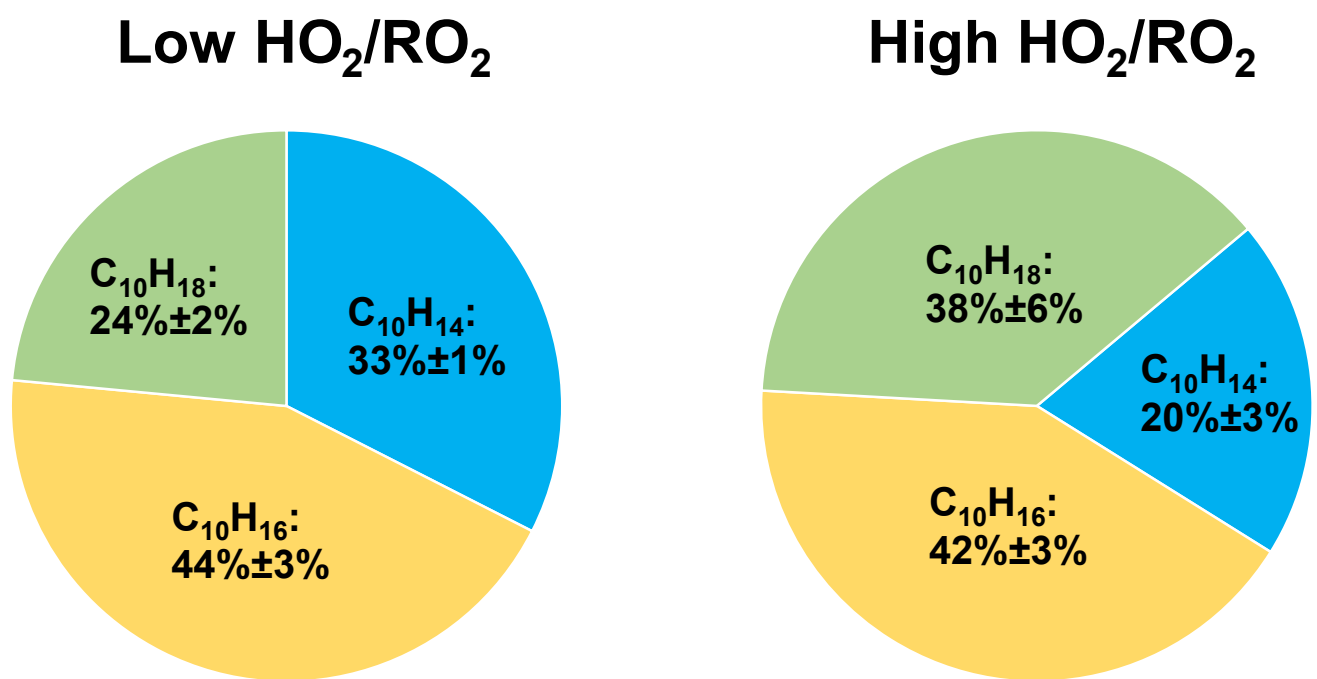

**Figure 11: Average contribution of the $C_{10}H_{14}O_z$, $C_{10}H_{16}O_z$, and $C_{10}H_{18}O_z$ family to the monomer class signal at low and high $HO_2/RO_2$ (pure gas phase experiments).**

The contribution of $C_{10}H_{16}O_z$ is largest and remains similar in both cases, matching the already shown unchanged signal level in **Fig. 6**. This is the case because the $C_{10}H_{16}O_z$ family contains the alcohols from $C_{10}H_{15}O_x$+$RO_2$, carbonyls from $C_{10}H_{17}O_x$+$RO_2$ and hydroperoxides from $C_{10}H_{15}O_x$+$HO_2$ (see **Fig. 1**). A separation of the effects of enhanced $HO_2$ on this monomer family is difficult, as for the case where $RO_2$ termination dominates vs. the case where $HO_2$ termination dominates, the loss of carbonyls and alcohols is partially compensated by the gain of hydroperoxides. A strong gain in hydroperoxides is clearly reflected in the strong increase of $C_{10}H_{18}O_z$ at high $HO_2/RO_2$.

Inspection of the $C_{10}H_{14}O_z$ and $C_{10}H_{18}O_z$ families shows that ~13 % of the contribution by $C_{10}H_{14}O_z$ are lost (carbonyls, 33 % at low $HO_2/RO_2$) and are present instead as $C_{10}H_{18}O_z$ (hydroperoxides), giving $C_{10}H_{18}O_z$ a contribution of 38 % at high $HO_2/RO_2$.

## 3.4 Impact on condensable organic mass

In the previous sections we demonstrated a shift of the product distribution by the shift from low to high $HO_2/RO_2$ conditions. We also showed that the changes could be rationalized by generic mechanistic considerations. We added $(NH_4)_2SO_4$ seed aerosol in two experiments to determine how the shift in the product distribution affects the condensable organic mass by determining the fraction which remained in the gas-phase after seeding.

**Figure 12** shows the fraction remaining for the sum of all products as well as for the individual product classes for the high and the low $HO_2/RO_2$ case. In both cases a significant reduction of products in the gas phase was observed with seed present. Overall, the sum of all products was reduced by around 60 %, with a slightly higher reduction in the low $HO_2/RO_2$ case. This can be attributed to the larger importance of HOM-Acc in the low $HO_2/RO_2$ case, as well as to a 10 % lower reduction of the HOM-Frag in the high $HO_2/RO_2$ case. In both cases a reduction of the HOM-$RO_2$ is observed, which indicates that the provided particle sink could have affected HOM formation chemistry, however only moderately.

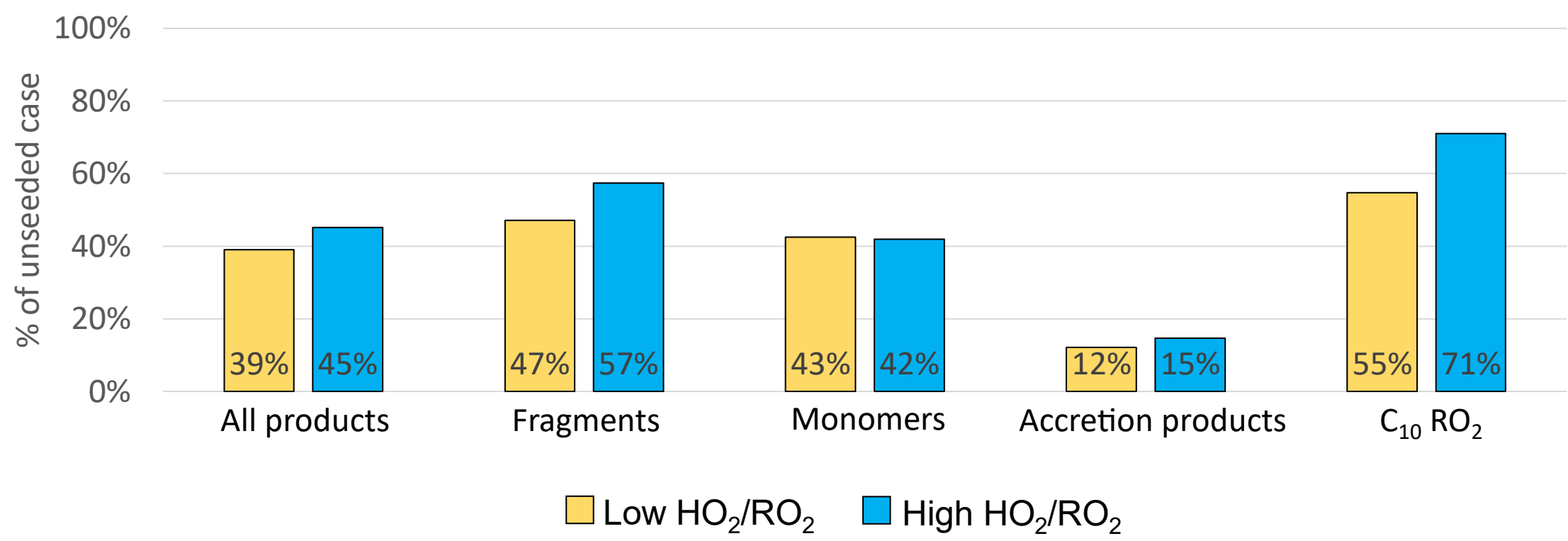

**Figure 12: Overview of average, relative change in product classes signal between gas phase only and seeded system. Blue shows the high $HO_2/RO_2$ case, yellow the low $HO_2/RO_2$ case. (All are normalized to α-pinene OH turnover, *Exp2* experiment)**

The total organic particulate mass was determined by AMS measurements and was 2.0 µg $m^{-3}$ and 3.4 µg $m^{-3}$ at high and low $HO_2/RO_2$ in the experiment (*Exp2*) displayed in **Fig. 12**. A reduction of condensed organic mass to 73 %±3 % at high $HO_2/RO_2$ (orange bar in **Fig. 14***)* was observed on average. Since non-seeded and seeded experiments were conducted at otherwise the same conditions and we did not observe significant new particle formation, the gas-phase compositions can be directly compared. Therefore, we conclude that the shift in the product distribution led to a reduction of condensable material at the same α-pinene turnover with OH (and $O_3$).

We calculated the wall loss corrected SOA yields with the corrected SOA mass as shown in **Eq. (4)** and as described by Sarrafzadeh et al. (2016). To this end we used $C_{10}H_{16}O_7$ as the lead HOM compound. In the two experiments with seed present (*Exp2.1* and *Exp3*) we had SOA yields of 7.3 % and 10.0 % at high $HO_2/RO_2$ and 10.0 % and 12.8 % at low $HO_2/RO_2$. The difference in the SOA yields between experiments can be explained by the slightly different OH concentrations and subsequent difference in contribution by photooxidation (see **Table 1**). Overall, our yields are in the lower range in comparison with the SOA yields reported by McFiggans et al. (2019) for the α-pinene photooxidation. However, our experiments were also performed at 5 °C higher temperature (20 °C compared to 15 °C in McFiggans et al. (2019)). The SOA yields show an absolute reduction of ~3 % at high $HO_2/RO_2$ compared to low $HO_2/RO_2$ (relative a reduction of about 30%). A reduction of the SOA yield of α-pinene by addition of CO was described before by McFiggans et al. (2019), however, there the α-pinene OH turnover was not held constant.

The change from low to high $HO_2/RO_2$ regime favored termination reactions to protic termination groups, as we observed less carbonyl compounds and more hydroperoxides. This could overall shift the product distribution to products with lower vapor pressures and favor SOA formation, since protic groups can act as hydrogen bond donors as well as hydrogen bond acceptors. (as exemplified by the comparison of ethanol (boiling point (b.p) 78 °C) and ethane hydroperoxide (b.p. 93-97 °C) with acetaldehyde (b.p. 20 °C) (Richter et al., 1955)). However, the effect of the termination group should be small for HOM as they likely contain multiple hydroperoxide groups (compare Pullinen et al. (2020)). The reduction in HOM-Acc is

expected to decrease the condensable mass, since the HOM-Acc scavenge non-HOM-$RO_2$, that would otherwise not partition into the particle phase.

Which of the measured compounds contribute significantly to the organic particle mass can be inferred by comparing their signal from the pure gas phase cases to their signal with seed in the system. Under the assumptions that, for most HOM-compounds re-evaporation to the gas phase is negligible and that the precursor chemistry is not substantially disturbed by seed addition, the fraction of signal remaining with seed in the system reflects to which degree the compound is condensing. **Figure 13** shows the fraction remaining with seed in the system plotted against the molar mass of each individual compound. The plot includes all closed shell products that were measured with a relative standard deviation of less than 30% for all measurement phases and depicts the results for both the high and low $HO_2/RO_2$ case.

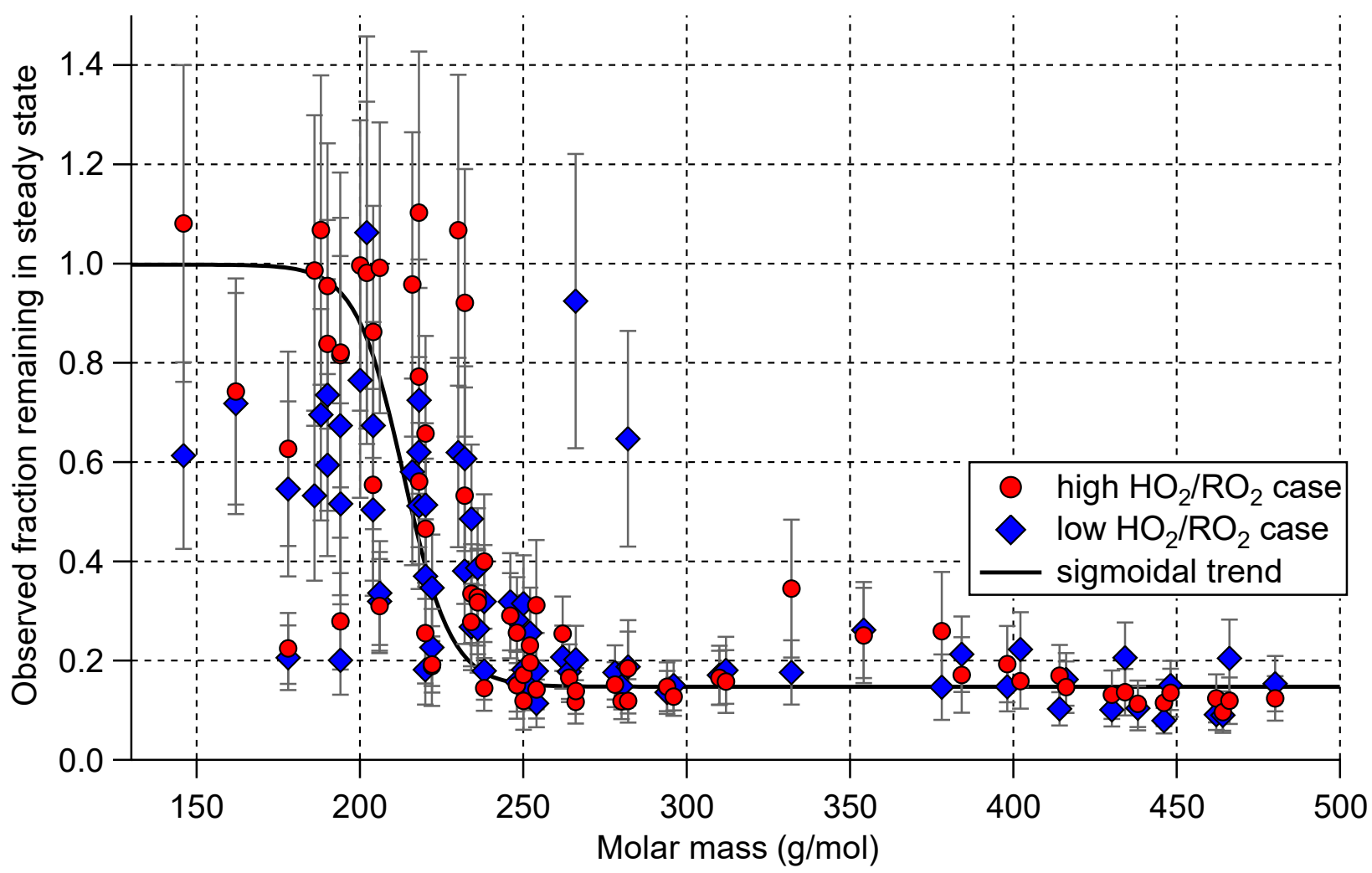

**Figure 13: Gas-phase fraction remaining in presence of seed (normalization of all data with α-pinene OH turnover) for the low (blue) and high (red) $HO_2/RO_2$ case. Displayed points represent all closed-shell compounds that were detected with relative standard deviation <30 % in all four experiment phases. Error bars represent result of error propagation (see supplement Sect. S7)**

Overall, in both cases we observed the same trend. Lighter compounds are not affected by the presence of seed particles, but with increasing molar mass the fraction remaining in the gas phase is reduced. A difference between the low and high $HO_2/RO_2$ case can be observed in the low molar mass range: In the high $HO_2/RO_2$ case many fragmentation products show a higher gas-phase fraction remaining up to 1. (In some cases, values larger than 1 were observed, however within the error limits. For the error estimation see supplement **Sect. S7**). Fractions remaining larger than 1 beyond error could be an indication that such products have a particle-phase production source. **Figure 13** also shows a critical SVOC/LVOC region for molar masses between 175 g $mol^{-1}$ and 250 g $mol^{-1}$ where neither a fraction remaining of 1 nor complete condensation is observed. The position of this region on the molar mass scale depends on the provided organic mass concentration. The large variation of the fraction remaining in this small range of molar masses shows that the partitioning coefficients are dependent

on the detailed structure of the compounds and not simply on their molar mass. The semi-volatile and low volatility products represent mainly higher oxidized fragments and HOM-Mon with less than 8 oxygen.

For compounds with a molar mass larger than 250 g $mol^{-1}$ a constant fraction remaining is reached in steady state, which is due to an ongoing production of the compounds. From the condensation behavior shown in **Fig. 13** we conclude that the compounds heavier than 230 g $mol^{-1}$ are expected to be of sufficiently low volatility to be mainly found in the particle phase for the organic mass present in the system and therefore contribute significantly to the SOA mass formation. Our finding agrees with the threshold used for low volatility HOM products in Pullinen et al. (2020).

Therefore, the signal of all compounds with a molar mass heavier than 230 g $mol^{-1}$ was weighted with their molar mass and summed (see **Eq. (2)**). The ratio of this weighted signal sum at low and high $HO_2/RO_2$ is then a measure of expected SOA mass loss. The calculation leads to an expected reduction of 72 % (blue bar, **Fig. 14***)*. This simplified approach leads to a good agreement with the AMS measurements and can thus explain the reduced particulate organic mass within the errors.

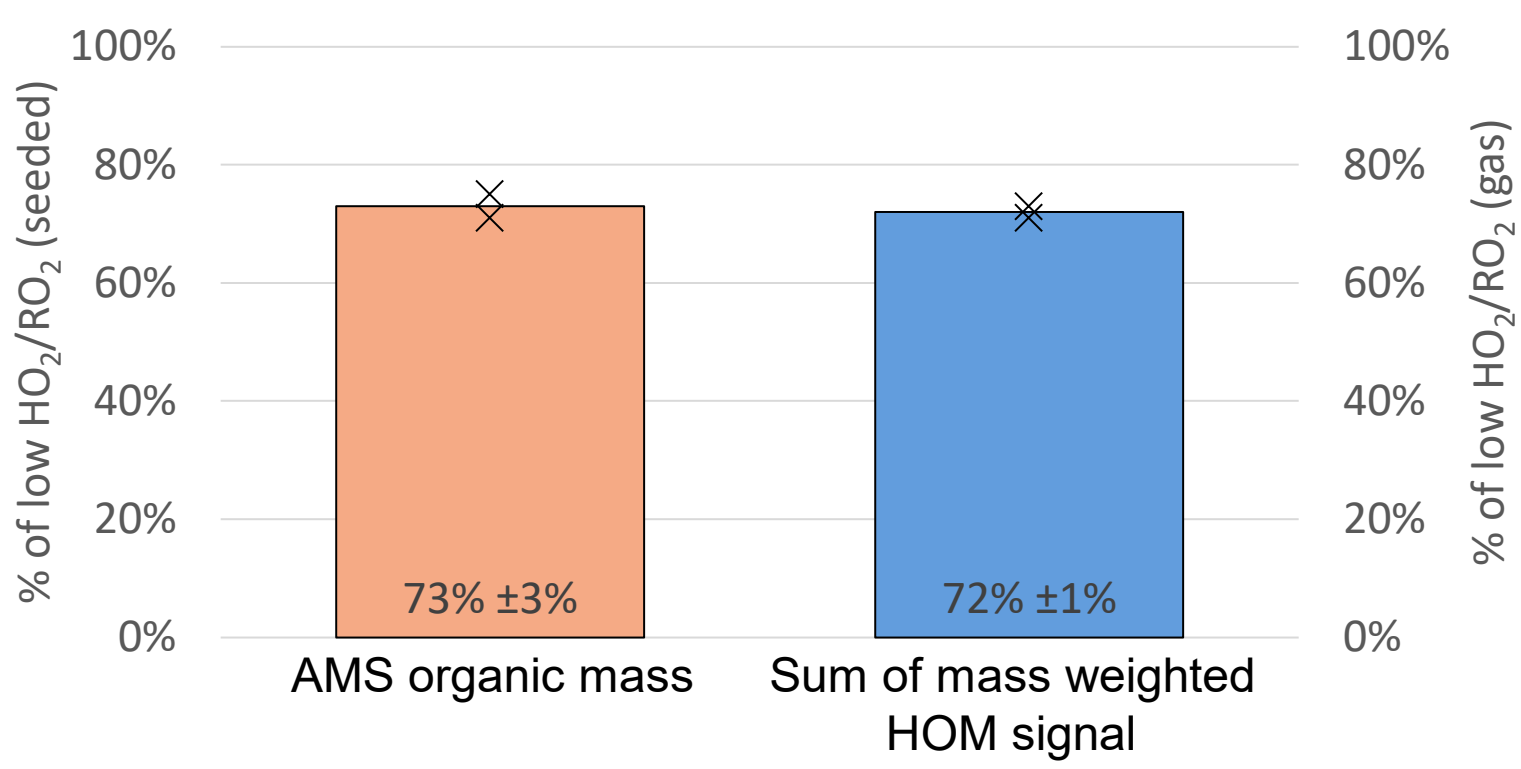

**Figure 14: Overview of the average, relative change in organic mass observed in the AMS (left y-axis, seeded experiments) and the mass weighted HOM signal observed in the $NO_3$-CIMS (right y-axis, pure gas phase experiments) between the low and high $HO_2/RO_2$ case (both normalized to α-pinene OH turnover).**

To test for closure between HOM lost and particulate organic mass measured we approximated the upper limit of HOM concentration in the condensed phase. For this calculation we used the calibration factor determined for sulfuric acid for our $NO_3$-MION-CIMS ($7.0 \cdot 10^9$ molecules$\cdot$ $cm^{-3} \cdot ncps^{-1}$) and the relationship between gas and particulate concentration of a compound in the SAPHIR STAR chamber described in **Eq. (3)**. Again, we considered all compounds with M>230 g $mol^{-1}$ in our calculation. The summed mass concentration lost from the gas phase was then compared to the SOA mass measured in the AMS. This comparison yields a good agreement within the uncertainties. The detailed calculation results can be found in the supplement (**Fig. S3**). Overall, an agreement within 40 % is achieved for all measurement stages.

The comparisons presented above show that we understand the processes governing the SOA formation in our chamber and that the $NO_3$-CIMS measurements are well suited to observe the critical changes to understand the reduction in condensable organic material when shifting from low to high $HO_2/RO_2$.

## 4 Conclusion

In the presented series of experiments, we achieved a shift from a $RO_2$+$RO_2$ dominated chemistry to a more atmospherically relevant $HO_2/RO_2$ ratio under constant α-pinene OH turnover. It was shown that moving towards atmospheric $HO_2/RO_2$ ratio affected the SOA formation potential, with the observed organic mass being reduced at high $HO_2/RO_2$. This is in support of the potential bias towards high SOA yields in chamber studies at low $HO_2/RO_2$ as discussed by Schervish and Donahue (2021). The gas-phase observations showed that the SOA reduction at high $HO_2/RO_2$ was mainly due to a reduced HOM-Acc formation which were formed by $RO_2$+$RO_2$ cross reactions in the low $HO_2/RO_2$ cases. This prevented contribution to SOA by less oxidized $RO_2$ which were scavenged in the HOM-Acc at low $HO_2/RO_2$. Under atmospheric condition such cross reactions are less important, and such (mixed) accretion products would contribute less to SOA.

The overall observed HOM-products were reduced slightly, showing that under certain circumstances $RO_2$+$HO_2$ termination can impede the HOM formation, mainly by reducing the precursor $RO_2$ levels and less by impeding the autoxidation itself. The autoxidation chain (once initiated) runs to a similar oxidation level at both high and low $HO_2/RO_2$. The observed HOM-Mon products shift significantly between monomer families due to the different termination reaction. A decrease in carbonyl and alcohol formation from $RO_2$+$RO_2$ and an increase in hydroperoxide formation from $RO_2$+$HO_2$ was observed at high $HO_2/RO_2$.

Furthermore, a reduction in HOM-Frag products, especially with lower carbon numbers, as well as the parity of the $C_{10}H_{15}O_x$ HOM-$RO_2$ show a reduction in alkoxy radical formation at high $HO_2/RO_2$. The moderate reduction in larger HOM-Frag products and pinonaldehyde, however, suggest that some alkoxy radical steps are still important. This raises the question of whether alkoxy radical formation can be facilitated by $HO_2$. In the atmosphere such effects are most often overcome whenever $RO_2$+NO is the major alkoxy radical source.

Overall, the observed changes in the gas phase could be well explained with the presented generic mechanistic understanding of HOM formation in the α-pinene system. The addition of seed demonstrated that the shift towards high $HO_2/RO_2$ reduced the condensable organic mass, stressing the importance of controlling higher order reactions of peroxy radicals which lead to overemphasis of HOM-Acc product formation at low $HO_2/RO_2$ ratios.

Furthermore, the seed addition allowed us to determine which products were contributing to the SOA formation and show that their volatility is a function of molar mass and detailed molecular structure. This revealed a critical mass region in which compounds have significant fractions in gas and particulate phase. Based on absorptive partitioning theory the volatilities at which this critical region is found should depend on the organic mass present in the system.

Valuable insight about the condensed phase can be gained from HOM gas phase measurements. We inferred conclusions about the particulate phase from the gas phase measurements and compared them to the direct particle phase observations, finding good agreements between our expectations and the measurements.

**Data availability**

All presented data will be available in a repository before the submission of the final manuscript.

**Author contribution**

TFM, MH and GM conceptualized the study and TFM, YB, SK and SRZ designed the experiments and developed the analysis methodology. The experiments were performed by YB, SK, VG and SRZ. Instrument deployment and/or data analysis were performed by YB, SK, HW, RW, JX, AZ, QH, TZ and VG. YB did model calculations of the experiments. AV, SPO, TJB, MG and MH provided counsel on experiment design and data interpretation. The compiled data set was interpreted by YB and TFM, and the results were discussed by all co-authors. YB visualized the data and YB and TFM prepared the manuscript. All co-authors reviewed the manuscript.

**Competing interests**

The authors declare that they have no conflict of interest.

**Financial support**

This research has received funding from the European Union’s Horizon 2020 research and innovation programme under the FORCeS grant agreement No 821205, the Federal Ministry of Education and Research (BMBF) Germany under the FONA Strategy “Research for Sustainability” as part of the implementation of ACTRIS-D under the funding code 01LK200010, Vetenskapsrådet (VR, grant agreement No. 2018-04430), Svenska Forskningsrådet Formas (grant agreement No. 2019-586) and the Natural Environment Research Council (NERC) UK under the grant agreement No. NE/V012665/1.

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
