# Peer review of "photooxidation products and secondary organic aerosol formation"

_EGUsphere, 2023_

## Author Comment (AC2)

This work describes experiments aimed at understanding how the chemistry of alpha-pinene oxidation changes under different HO2:RO2 conditions. As described in the paper, this is crucial to understand as high precursor concentrations and low concentrations of small RO2 molecules in chamber experiments lead to lower HO2:RO2 than is generally expected in the real atmosphere. While this is speculated to lead to lower SOA yields, this work shows that as well as evidence for the chemistry shifting when RO2 versus HO2 dominate. Additionally the authors show an interesting method for estimating SOA yields from a CIMS measurement of large gas-phase molecules. This work brings attention to a limitation of interpreting many chamber experiments that show dominant RO2-RO2 chemistry to the real atmosphere. Additionally, this work is thorough in describing their results in the context of known alpha-pinene chemistry and is clear about the limitations in their interpretation. I feel this work fits well within the scope of ACP and I recommend publication once a few issues have been addressed.

General comments

1. I am a little confused about Exp2. Specifically what is the importance of doing these consecutively versus having two separate experiments (one seeded, one unseeded)? What is the difference between an unseeded experiment and a pure gas phase experiment?
2. A little more discussion of the atmospheric relevance of the conditions would be helpful. Where in the atmosphere is an HO2:RO2 of 1 relevant? As both of these values vary across the globe are there areas where a lower or higher ratio would in fact be more representative?
3. A little more discussion of how this impacts how chamber experiments should be run could also be useful. Can you reference any typical HO2:RO2 for chamber experiments (even just ones inferred from modeling specific chamber conditions)? Besides Schervish and Donahue (2021), which was exclusively a modeling paper not aimed at specifically reproducing any experiment, is there evidence for non-atmospherically relevant ratios in chamber experiments? Would a simple adjustment of yields be sufficient to account for low HO2:RO2 in chambers? Would a model such as the box model with MCM that you used allow these experiments to be interpreted at higher HO2:RO2? Or are experiments that intentionally increase this ratio necessary?
4. The model results are referenced a few times, but never shown. These should at least be included in the SI whenever they are mentioned in the text. Additionally a discrepancy between the modeled and measured reduction in C10-HOM-RO2 is mentioned, but is never appropriately explained. Was any other comparison done between specific product or RO2 families and the model results besides just total C10-HOM-RO2? It seems like that could provide more evidence for some of the interpretations made here (for example for the results in Fig 7).

Specific comments:

1. Line 26: A reduction relative to what
2. Line 66: This paragraph seems more appropriate in the methods section.

3. Line 91: This sentence is confusing. Suggests this process as what?
4. Line 92: The autoxidation rate for some RO2 may be in this range while others might be much slower or faster. Does this accurately represent the average?
5. Line 94: A reference should be provided to justify the autoxidation rate coefficient slowing down as more oxygen is added.
6. Line 238: I think you've mixed up these rate coefficients.
7. Line 262: By "desired value" do you mean the value it was before CO addition?
8. Line 283: The value of the RO2+RO2 rate coefficient has been shown to vary substantially based on the structure of the RO2's. Were any sensitivity studies done to see if the model was particularly sensitive to the value chosen?
9. Line 363: The discussion here is confusing to me. You do see a reduction in HOM-RO2 so why is this relevant to discuss?
10. Line 410: What RO2 ratio?
11. Line 472: Here it is stated that HOM-C10H17Ox are less abundant, but earlier (line 449) it is stated that it is expected there are the products of the dominant OH pathway despite being detected as lower in the NO3-CIMS. Is this why it is assumed there is an abundance of less oxidized C10H17Ox peroxy radicals? Are there any measurements to validate this?
12. Line 502-502: Wouldn't 1 alkoxy step lead to the same parity change? Why is it then suggested in both cases where a difference in the amount of parity change exists?
13. Line 636-637: What would cause there to be a larger particle-phase source of these compounds at high HO2:RO2?
14. Line 650: Should this be "reduction to 72%"?
15. There are a few compiling errors in the main text and SI. I would recommend going through and carefully checking all the equation/figure references are showing up correctly.
* * *
***ANSWER TO REVIEWER #1:***

We thank the referee for carefully reviewing our manuscript and for the insightful comments and the constructive feedback! Please find our answers to each individual point below:

General points:

1. "Pure gas phase" and "unseeded" can indeed be used interchangeably in our case since both refer to the experiments without particle addition and we did not observe significant particle formation. We highlighted in Section 3.2 Experiment Conditions (Line 254-255) that no nucleation was observed in the analysed steady states and changed all descriptions to uniformly refer to the particle-free experiments as "unseeded":

*L254-255: In the unseeded experiments no significant nucleation was observed leading to pure gas phase conditions.*

      Regarding Experiment 2 the referee is correct, one could separate the experiments completely, conducting a seeded and an unseeded one. However, it is of advantage to perform them in one run as this ensures that experimental parameters are the same in the seeded and unseeded case, which is (out of experience) otherwise not so simple to guarantee in complex setups as ours.

2. A paragraph about the relevance of different HO$_2$/RO$_2$ ratios was added in the result section (Line 352-356) including the addition of exemplary measurements of HO$_2$/RO$_2$ ratios in field campaigns under urban to remote atmospheric conditions in the supplement (**Table S5**) showing HO$_2$/RO$_2$ ratios under different conditions.

*L352-356: HO$_2$/RO$_2$ ratios of around 1 are highly relevant for atmospheric conditions with significant OH oxidation, though it should be kept in mind that in atmospheric conditions the methyl peroxy radical and other small RO$_2$ contribute a significant portion to the total of peroxy radicals (Khan et al., 2015). Field studies reporting HO$_2$ and RO$_2$ measurements for different environments can be found in supplement Table S5. These exemplary studies show that HO$_2$/RO$_2$ ratios around 1 are relevant in remote to urban environments with different VOC sources and NO$_x$ levels.*

3. Thanks to the referee for this helpful comment. Two publications discussing the importance of the HO$_2$/RO$_2$ ratio for SOA yield determination were added in Lines 59-60:

*L59-60: Previous studies of VOC ozonolysis with different OH scavengers by Docherty and Ziemann (2003) and Keywood et al. (2004), indicated a significant impact of the HO$_2$/RO$_2$ ratio on SOA yields.*

HO$_2$/RO$_2$ ratios are not very often reported, as these species are difficult to measure directly. However, as pointed out by Schervish and Donahue (2021), as well as by Bianchi et al. (2019) and Bell et al. (2023) in experiments with only a-pinene/VOC and oxidant, absence of HO$_2$ production channels in such systems will lead inevitable to low HO$_2$/RO$_2$ ratios. Bell et al. (2023) calculated the HO$_2$/RO$_2$ ratios in their α-pinene ozonolysis experiments regarding in presence of different OH scavengers and got an HO$_2$/RO$_2$ ratio in the range of 1E-4 in the absence of OH scavengers.

To the second part of the question regarding an adjustment factor from low to high HO$_2$/RO$_2$ for SOA yields: As the processes (especially accretion product formation) are not linear it is not straight forward to extrapolate from low to high HO$_2$/RO$_2$.
Unfortunately, the chemistry in the MCM does not include HOM or HOM-accretion product formation and can therefore not be used to infer the effect on HOM formation at higher HO$_2$/RO$_2$. It can, however, yield general expectations for the HO$_2$ and RO$_2$ concentration to design experiments accordingly.
In the presented experiments we try to isolate the HO$_2$/RO$_2$ ratio as one important parameter to understand the importance of the termination of (HOM-)RO$_2$ radicals by HO$_2$ and RO$_2$ via shifting HO$_2$/RO$_2$ towards a ratio more relevant for the atmosphere. Another very important factor is the termination reaction of RO$_2$ with NO. (This will be addressed in our next paper.) A paragraph highlighting the importance of considering the different RO2 sinks was added in the conclusion (Line 691-693):

*L691-693: Our results confirm that too low HO$_2$/RO$_2$ is one important parameter that can lead to an overestimated SOA yield in laboratory studies. In a broader picture the results show how important it is to consider the different contributions to the HOM-RO$_2$ sink (e.g. HO$_2$, RO$_2$. NO) when designing experiments and transferring laboratory results to the real atmosphere.*

4. Model results are now included in the supplement Section S4. We cannot compare the measured HOM-RO$_2$ with the modelled RO$_2$ as the MCM does not represent any HOM chemistry. The discrepancy between the modelled RO$_2$ ***sum*** and the observed HOM-RO$_2$ is

due to the constant production of the first generation $RO_2$ (by controlling the α-pinene OH turnover) while the chemical sink increases at higher $HO_2/RO_2$. The chemical sink for an individual $RO_2$ via autoxidation is missing in the MCM. Under the assumption that the autoxidation outcompetes the termination reactions in many cases this leads to a lower $RO_2$ reduction than what is expected in the model for $RO_2$ that undergo autoxidation. A sentence to clarify this explanation was added in Line 456-458:

*L456-458: This consideration shows that the smaller reduction in HOM-RO$_2$ compared to the lower oxidized RO$_2$ in the model is compatible with fast autoxidation reactions that are missing in the MCM.*

Specific comments:

1. Specified the reduction of the HOM's SOA formation potential in comparison to low $HO_2/RO_2$ (Line 26-27):

*L26-27: We determined a reduction of the HOM's SOA formation potential by ≈30 % at HO$_2$/RO$_2$≈1/1 compared to HO$_2$/RO$_2$≈1/100.*

2. We agree and moved the contents of this paragraph down to Section 2.2 (Methods, Control of a-pinene turnover) and replaced the explanation with a more concise description of the experimental concept in Line 67-72 instead of going into details already.

*L67-72: We compared two experimental conditions, a pure α-pinene photooxidation case leading to low HO$_2$/RO$_2$ ratios and high importance of RO$_2$ cross-reactions and a high HO$_2$/RO$_2$ case representing more atmospheric relevant conditions with high importance of RO$_2$+HO$_2$ reactions. One important concept of the conducted experiments is the constant OH availability to α-pinene in order to prevent effects of different oxidant levels and allow for a direct comparison between the two chemical regimes. To this end, the OH concentration in the experiments was adjusted to keep the α-pinene OH turnover constant and to avoid changes due to oxidant scavenging.*

3. Clarified the sentence by adding that the direct H-abstraction from α-pinene is suggested as the start point of the autoxidation chain (Line 91-92):

*L91-92: A recent study suggests direct H-abstraction by OH from α-pinene (Shen et al., 2022) as a starting point for the autoxidation chain.*

4. It is difficult to give a good estimation of an average autoxidation rate as the rate depends strongly on the specific structure of the peroxy radical. For this reason, we can only give a range of published autoxidation rates (see also next answer to point 5).

5. This sentence was rephrased to be clearer (e.g. not the oxidation degree per se defines the rate of possible H-shifts, but the availability of H-atoms in favourable positions) and two citations were added (Line 94-97):

*L94-97: The autoxidation chain will run quickly, adding more oxygen to the molecule, until bimolecular termination reactions are able to compete with all available H-shift rates. The rate of an H-shift is determined by the hydrogen's position in relation to the peroxy radical and the functional groups near the hydrogen and peroxy radical (Otkjaer et al., 2018; Vereecken and Nozière, 2020).*

6. Correct! We switched the rate constants.

7. The desired value for the OH concentration is indeed the same as before the CO addition. We clarified this (Line 271-272).

*L271-272: In the displayed Exp1 the OH level was adjusted in three steps to approach the same concentration as before the CO addition.*

8. The stated $RO_2+RO_2$ reaction rate constant was used in the considerations regarding the sink contributions and expected reduction of $RO_2$ concentrations. For these calculations we tested different reaction rate constants, and the results are described in Section 4.2 (Line 428-437). The $RO_2+RO_2$ reaction rate of $k_{RO2+RO2}= 5\cdot10^{-12}$ $cm^3\cdot s^{-1}$ was not applied in the modelling, there the reaction rates provided by the MCM were used.

9. The mentioned paragraph deals with the question why the HOM-Mon products don't show a reduction even though the precursor $RO_2$ are decreased and aims to show that this is due to reduced HOM-Acc formation. We agree that this was not ordered well and modified the script accordingly (Line 379-387).

*L379-387: Without changes in the rates and contributions of the different termination reactions, the observed reduction in the HOM-RO₂ precursors should lead to nearly the same reduction in HOM-Mon. However, the decrease of accretion product formation and fragmentation should lead to an increase in HOM-Mon. The presence of HO₂ could reduce the alkoxy formation, and thus fragmentation of HOM-RO₂. This missing sink could lead to an additional HOM-Mon source compared to the low HO₂/RO₂ case. However, the distribution of the product classes at low and high HO₂/RO₂ (**Fig. 5**) shows that contributions are shifted from HOM-Acc to HOM-Mon, while the contribution of HOM-Frag remains constant. Each HOM-Acc is formed from one HOM-RO₂ (HOM-RO₂+RO₂) or potentially even two HOM-RO₂ (HOM-RO₂+HOM-RO₂) and therefore each HOM-Acc not formed will lead to at least one HOM-Mon.*

10. We restructured the sentence to make clearer that we are comparing the expected $RO_2$ concentration at high and low $HO_2/RO_2$ (Line 428-430).

*L428-430: For steady state conditions, we can estimate the expected effect on the RO₂ ratio between high and low HO₂/RO₂ conditions for those HOM-RO₂ with production directly linked to the primary production ($k_{OH}\cdot[OH]\cdot[α\text{-pinene}]$) with negligible further autoxidation.*

11. $C_{10}H_{17}O_X$-$RO_2$ are the main expected products from a-pinene+OH and this assumption is supported by our findings for the contribution of different HOM-Acc families, as the high contribution of $C_{20}H_{32}O_z$ but the small contribution of $C_{10}H_{17}O_X$ HOM-$RO_2$ can be explained by a high abundance of lower oxidized $C_{10}H_{17}O_X$. Berndt (2021) recently reported measurements of these lowly oxidized $C_{10}H_{17}O_X$-$RO_2$ using a CI-MS with ethylaminium as the reagent ion. A reference to this source was added in Line 491:

*L491: Lower oxidized C₁₀H₁₇Oₓ-RO₂ were recently measured by Berndt (2021).*

12. We focussed Line 521-524 and indicate now only that in the simplest case at low $HO_2/RO_2$ one alkoxy step takes place, and at high $HO_2/RO_2$ none takes place.

*L521-524: In the simplest case 1 alkoxy step takes place at low HO₂/RO₂ due to HOM-RO formation from HOM-RO₂+RO₂ reactions, while no alkoxy step takes place at high HO₂/RO₂, because HOM-RO₂+HO₂ produces none or less HOM-RO than HOM-RO₂+RO₂.*

13. As the values for fraction remaining larger than 1 are well within the error estimation, we don't expect a major particle source of these compounds, though we cannot exclude the possibility. Here more investigation would be necessary.

An explanation for the observation of more compounds with a fraction remaining larger than 1 in the high $HO_2/RO_2$ case could be the larger importance of higher volatility products in this mass range, as we observed a larger importance of higher oxidized HOM-fragments in low $HO_2/RO_2$ conditions: Contribution of fragments with O/C>1 was 25 % at high $HO_2/RO_2$ and 35 % in the low $HO_2/RO_2$ case. This is also reflected in the lower reduction of HOM-fragments in the presence of seed at high $HO_2/RO_2$ in Figure 12.

14. Yes, this should say "to 72 %" and was changed in the manuscript (Line 670).

*L670: The calculation leads to an expected reduction to 72 % (blue bar, **Fig. 14**).*

15. Thanks for pointing this out, in-text references have been checked and fixed.

References:

Bell, D. M., Pospisilova, V., Lopez-Hilfiker, F., Bertrand, A., Xiao, M., Zhou, X., Huang, W., Wang, D. S., Lee, C. P., Dommen, J., Baltensperger, U., Prevot, A. S. H., El Haddad, I., and Slowik, J. G.: Effect of OH scavengers on the chemical composition of α-pinene secondary organic aerosol, Environ. Sci. Atmos., 3, 115-123, https://doi.org/10.1039/d2ea00105e, 2023.

Berndt, T.: Peroxy Radical Processes and Product Formation in the OH Radical-Initiated Oxidation of α-Pinene for Near-Atmospheric Conditions, J. Phys. Chem. A, 125, 9151-9160, https://doi.org/10.1021/acs.jpca.1c05576, 2021.

Bianchi, F., Kurtén, T., Riva, M., Mohr, C., Rissanen, M. P., Roldin, P., Berndt, T., Crounse, J. D., Wennberg, P. O., Mentel, T. F., Wildt, J., Junninen, H., Jokinen, T., Kulmala, M., Worsnop, D. R., Thornton, J. A., Donahue, N., Kjaergaard, H. G., and Ehn, M.: Highly Oxygenated Organic Molecules (HOM) from Gas-Phase Autoxidation Involving Peroxy Radicals: A Key Contributor to Atmospheric Aerosol, Chem. Rev., 119, 3472-3509, https://doi.org/10.1021/acs.chemrev.8b00395, 2019.

Docherty, K. S. and Ziemann, P. J.: Effects of stabilized criegee intermediate and OH radical scavengers on aerosol formation from reactions of β-pinene with O3, Aerosol Sci. Tech., 37, 877-891, https://doi.org/10.1080/02786820300930, 2003.

Keywood, M., Kroll, J., Varutbangkul, V., Bahreini, R., Flagan, R., and Seinfeld, J.: Secondary organic aerosol formation from cyclohexene ozonolysis: Effect of OH scavenger and the role of radical chemistry, Environ. Sci. Technol., 38, 3343-3350, https://doi.org/10.1021/es049725j, 2004.

Khan, M., Cooke, M., Utembe, S., Archibald, A., Derwent, R., Jenkin, M. E., Morris, W., South, N., Hansen, J., Francisco, J., Percival, C. J., and Shallcross, D. E.: Global analysis of peroxy radicals and peroxy radical-water complexation using the STOCHEM-CRI global chemistry and transport model, Atmospheric Environment, 106, 278-287, https://doi.org/10.1016/j.atmosenv.2015.02.020, 2015.

Otkjaer, R. V., Jakobsen, H. H., Tram, C. M., and Kjaergaard, H. G.: Calculated Hydrogen Shift Rate Constants in Substituted Alkyl Peroxy Radicals, J. Phys. Chem. A, 122, 8665-8673, https://doi.org/10.1021/acs.jpca.8b06223, 2018.

Schervish, M. and Donahue, N. M.: Peroxy radical kinetics and new particle formation, Environ. Sci. Atmos., 1, 79-92, https://doi.org/10.1039/d0ea00017e, 2021.

Shen, H., Vereecken, L., Kang, S., Pullinen, I., Fuchs, H., Zhao, D., and Mentel, T. F.: Unexpected significance of a minor reaction pathway in daytime formation of biogenic highly oxygenated organic compounds, Sci. Adv., 8, eabp8702, https://doi.org/10.1126/sciadv.abp8702, 2022.

Vereecken, L. and Nozière, B.: H migration in peroxy radicals under atmospheric conditions, Atmos. Chem. Phys., 20, 7429-7458, https://doi.org/10.5194/acp-20-7429-2020, 2020.

---

## Author Comment (AC3)

Summary and general comment:

Baker et al. present very interesting results obtained from laboratory experiments done at the SAPHIR STAR chamber based in Julich. In these experiments the authors have analysed the HOMs formation using a-pinene as precursors and their impact on SOA. In these experiments they focus on two different main conditions: High and low $HO_2/RO_2$ ratio. This study is very important because up to now most of the laboratory experiments have been done at relatively low $HO_2/RO_2$ ratio however, if we want to mimic atmospheric daytime conditions, we need experiments also at high $HO_2/RO_2$ ratio. In these experiments, they have found that at high ratio they observed a decrease of HOM-accretion products and several other changes. Basically, as the authors mentioned, the study showed that low HO2/RO2 ratios can lead to an overestimation of SOA yields. Because of all these findings, I believe that this article is suitable for publication in ACP. Below I have added few minor comments.

**Minor comments:**

Line 22-24 Rephrase the sentences:*The HO2/RO2 ratio was increased by adding CO, while keeping the OH concentration constant. We determined the HOM's SOA formation potential, considering their fraction remaining in the gas phase after seeding with (NH4)2SO4 aerosol.*"

I think that these sentences lack of details and probably don't belong in the abstract. I would either explain the details of how the experiments were done or simply mention the high and low ratio and remove them from the abstract.

*Introduction*

The introduction is very well written, and it includes a nice set of references. My only minor comment here is related to the final sentences at line 79-80. The authors mentioned the used of mass spectrometry with chemical ionization (HR-TOF-CIMS). This is a very general term where these techniques can include a very large set of instruments and different ionization unit. Would be nice if the author can explain better with techniques they used and why.

*Method*

The method part is also very clear. However, I have one main comment. I might have missed that but how many experiments and how many repetition were done? While reading the paper I had the feeling that there were not many repetitions for each experiment type. I am aware that those experiments requite lots of work and lots of data analysis. I was just wondering how representative they are and would be the impact on their conclusions.

*Results and Discussions*

The authors include a vast description of their results. One comment here is about figure 4 and its discussion. I understand very well that the accretion products decrease. However, I would expect an increase in the monomers. The authors mentioned that shortly, but I believe more discussions is needed. For example, can it be due to the limited numbers of experiments and relatively low statistics? Is there any other explanation? Maybe is due to the normalization the authors used? As motioned above the rest of the discussions is very clear and full of details.

***ANSWER TO REVIEWER #2:***

We thank the referee for carefully reviewing our manuscript and for the constructive feedback. Please find our answers to the individual points below:

Comment on Abstract:

We understand the principal concerns of the referee. However, we would like to mention that the results could be only gained by applying two relative new concepts:
1. Readjusting the OH concentration such that the primary turnover is nearly the same and effects arising from the different OH concentrations are suppressed.
2. We seed virtually the same evolved chemical system at steady state and use the fraction remaining to directly "measure" the SOA formation potential.
These "novelities" are conceptually inherent to our experiments and seemed important enough to be mentioned as a kind of a teaser in the abstract. However, the abstract would become lengthy when giving details beyond a teasing. They will be explained later in more detail.

Comment on Introduction:

A sentence was added in Line 78-80 to highlight the use of NO3-CIMS due to its high selectivity for HOM compounds.

*L78-80: As the central analysis tool, we will use high resolution time of flight mass spectrometry with chemical ionization (HR-TOF-CIMS) with nitrate ($NO_3^-$) reagent ions as this ionization scheme is selective towards HOM compounds (Hyttinen et al., 2018).*

Comment on Method:

Four experiments were performed, two unseeded experiments only investigating the gas phase and two experiments with addition of ammonium sulfate seed aerosol. Therefore, one repetition of each condition was conducted and a sentence clarifying this was added in Lines 251-252. The individual experiments as well as the average are represented in the bar graphs in the Results and Discussion section. The results for the individual experiments are represented by the markers, showing that the results were well reproduced between experiments. The same was observed for the experiments with seed compared to the unseeded experiments. We would like to mention that missing repetition is in parts compensated by multi-instrumental approach, which allows for consistency checks. Of course, more repetitions would give a better statistical basis.

*L251-252: Four experiments were performed in total, leading to one repetition of each studied condition.*

Comment on Results and discussion:

As pointed out a decrease in accretion products should lead to an increase in monomers, as the HOM-$RO_2$ that terminated as accretion products at low $HO_2/RO_2$ will terminate to monomer products (hydroperoxide termination group) with $HO_2$.
However, we additionally observed a reduced steady state concentration of HOM-$RO_2$ themselves. This reduction can be due to a faster chemical sink at high $HO_2/RO_2$, but also a reduced production of HOM-$RO_2$. As explained in Section 4.2 (Lines 438-443) a missing source term is expected for $C_{10}H_{15}O_x$ due to its formation pathways including alkoxy radical formation. This missing source term can explain that no overall increase in HOM-monomers was observed at high compared to low $HO_2/RO_2$.

A more in-depth explanation of the expected changes and shifts between product classes was added in Line 379-387:

*L379-387: Without changes in the rates and contributions of the different termination reactions, the observed reduction in the HOM-RO$_2$ precursors should lead to nearly the same reduction in HOM-Mon. However, the decrease of accretion product formation and fragmentation should lead to an increase in HOM-Mon. The presence of HO$_2$ could reduce the alkoxy formation, and thus fragmentation of HOM-RO$_2$. This missing sink could lead to an additional HOM-Mon source compared to the low HO$_2$/RO$_2$ case. However, the distribution of the product classes at low and high HO$_2$/RO$_2$ (**Fig. 5**) shows that contributions are shifted from HOM-Acc to HOM-Mon, while the contribution of HOM-Frag remains constant. Each HOM-Acc is formed from one HOM-RO$_2$ (HOM-RO$_2$+RO$_2$) or potentially even two HOM-RO$_2$ (HOM-RO$_2$+HOM-RO$_2$) and therefore each HOM-Acc not formed will lead to at least one HOM-Mon.*